# Physics-Informed Autoencoder for Enhancing Data Quality to Improve the Forecasting Reliability of Cabon Dioxide Emissions from Agricultural Fields

## Abstract

Missing values in measurements for carbon dioxide emissions on drained peatlands remains an open challenge for training forecasting techniques to achieve net zero. Existing methods struggle to model $CO_2$ emissions to fill gaps at the field scale, especially in nighttime measurements. We propose novel Physics-Informed Autoencoders (PIAEs) for stochastic differential equations (SDEs), which combine the generative capabilities of Autoencoders with the reliability of physical models of Net Ecosystem Exchange (NEE) that quantify $CO_2$ exchanges between the atmosphere and major carbon pools. Our method integrates an SDE describing the changes in NEE and associated uncertainties to fill gaps in the NEE measurements from eddy covariance (EC) flux towers. We define this SDE as a Wiener process with a deterministic drift term based on day and night time NEE physics models, and stochastic noise term. In the PIAE model, various sensor measurements are encoded into the latent space, and a set of deterministic decoders approximate the SDE parameters, and a probabilistic decoder predicts noise term. These are then used to predict the drift in NEE and thereby the optimal NEE forecast at the next time instance using the SDE. Finally, we use a loss function as a weighted sum of the Mean Squared Error (MSE) and Maximum Mean Discrepancy (MMD) between the measurements and the reconstructed samples and the associated noise and drift. PIAE outperforms the current state-of-the-art Random Forest Robust on predicting nighttime NEE measurements on various distribution-based and data-fitting metrics. We present a significant improvement in capturing temporal trends in the NEE at daily, weekly, monthly and quarterly scales.

## 1 Introduction

Greenhouse gas (GHG) emissions and removals can be monitored at various local and country-wide levels. At the local scale, flux towers using eddy covariance (EC) systems measure Net Ecosystem Exchange (NEE), Latent Heat (LE), and Sensible Heat (H) among other atmospheric scalars Zhu et al. (2022). However, measurements often contain gaps due to power shortages, device malfunctions, or other issues, ranging from half-hourly to several months. Gap-filling methods are employed to enhance data quality for forecasting and analysis. Additional information can be obtained by incorporating complementary measures from other tools, such as satellite observations through remote sensing.

Initial gap-filling strategies leveraged flux covariance with meteorological variables and temporal auto-correlation, but they struggle with gaps longer than 12 days Reichstein et al. (2005). A comprehensive study by Moffat et al. (2007) evaluated 15 techniques across different gap scenarios, finding that Non-Linear Regression (NLR), Look-Up Table (LUT), Marginal Distribution Sampling (MDS), and the Semi-Parametric Model (SPM) performed well overall, though challenges remain for gaps up to one month.

To address longer gaps, the Random Forest Robust (RFR) method was introduced, improving R2 values for NEE by 30% compared to MDS and reducing uncertainty by 70% Zhu et al. (2022). While effective for longer gaps, RFR struggles with nighttime measurements.

Table 1: List of variables from the flux data Cumming et al. (2020)

| Variable | Units | Description |
|---|---|---|
| NEE | $\mu$mol C m$^{-2}$s$^{-1}$ | Net ecosystem $CO_2$ exchange flux density before data gap-filling |
| H | W. m$^{-2}$ | Sensible heat flux density |
| Tau | kg. m$^{-1}$.s$^{-2}$ | Momentum flux |
| RH | % | Relative humidity at 2m |
| VPD | HPa | Vapor pressure deficit |
| $R_g$ | W. m$^{-2}$ | Global radiation |
| Ustar | m. s$^{-1}$ | Friction velocity |
| Tsoil1 | °C | Soil temperature at depth of 0.05m |
| Tsoil2 | °C | Soil temperature at depth of 0.05m |
| $T_{air}$ | °C | Air temperature at 2m |

This research aims to build on gap duration agnostic approaches, improving performance and interpretability in modeling NEE measurements based on physical laws. We propose Physics-Informed Autoencoders (PIAE), which integrate NEE physics models for day and night with a stochastic differential equation (SDE) to predict NEE values and fill data gaps. Our method also provides forecasting capabilities and enhances performance on NEE gap-filling by accurately learning the NEE distribution and associated parameters.

In summary, PIAE combines empirical nighttime and daytime NEE models with a noise model White & Luo (2008) to improve robustness and accuracy in filling gaps and forecasting NEE. Our key contributions include:

- Introducing a stochastic differential equation for NEE measurements combining daytime and nighttime models with Gaussian noise.

- Introducing an interpretable PIAE model that accurately forecasts the NEE at next time stamps as a combination of deterministic drift and noise terms, guided by an SDE.

- Demonstrating that PIAE improves gap-filling robustness compared to state-of-the-art methods, handling gaps from months to years.

- Showing significant improvements in NEE distribution learning validated by better Maximum Mean Discrepancy (MMD), Wasserstein distance, and Kullback-Leibler (KL) divergence.

- Accurately predicting SDE parameters, enhancing interpretability.

The following sections introduce the flux tower dataset, detail the NEE models and associated noise, and demonstrate the robustness of the PIAE gap-filling method and its forecasting capabilities. Finally, we discuss limitations and potential improvements.

## 2 FLUX TOWER DATA

The measurements we use for this research are collected from the flux tower situated in East Anglia, UK Cumming et al. (2020), pictured in Figure 3. It collects several meteorological measurements including different flux entities such as NEE, sensible heat flux density (H) and latent heat (L) along with air and soil temperatures and radiation as well as timestamps. The subset of the data used in this research was collected for 8 years between 2012 and the end of 2019 every 30 minutes. Table 1 describes the variables in the data used in experimentation in this research. As the same instrument measure NEE and latent heat (L), if a value is missing for NEE this might be missing for L, thus we do not consider L in the experiments. In addition, we also include time-based attributes for each data point in the experimentation including season, hour, day of week, month and day of year.

## 3 Net Ecosystem Exchange Dynamics: Nighttime and Daytime Models

Since we consider physical measurements, we need to consider two aspects: the drift of NEE (deterministic part) and the noise of measurements (assumed to be Gaussian) White & Luo (2008). Thus, we introduce the dynamics of NEE over time $t$, modeled as a Stochastic Differential Equation (SDE) incorporating a Wiener process. Firstly, we define a simpler drift of the NEE model as a function of time-based on the Arrhenius-type law for NEE Lasslop et al. (2010) using the temperature and radiation models. We then introduce the diffusion coefficient to represent the noise and complete the SDE. Finally, the SDE is incorporated into the PIAE architecture (discussed in Section 4). The small number of parameters of these dynamics makes it easier to use in the architecture.

### 3.1 The Net Ecosystem Exchange nighttime and daytime models

The Net Ecosystem Exchange (NEE) represents the net exchanges of $CO_2$ between the ecosystem and the atmosphere Lasslop et al. (2010); Keenan et al. (2019). NEE accounts for ecosystem respiration, the release of $CO_2$ from biological activity, and photosynthesis, the uptake of $CO_2$ to produce energy for a plant. Thus, NEE is decomposed as the difference between ecosystem respiration ($R_{eco,t}$) and the Gross Primary Product (GPP). GPP represents photosynthesis, following the convention that negative fluxes indicate the removal of $CO_2$ from the atmosphere Lasslop et al. (2010), for every time $t$,

$$NEE_t = R_{eco,t} - GPP_t \tag{1}$$

where $R_{eco,t}$ and $GPP_t$ are parameterized by temperature and radiation values over time $t$. Since, the SDE models NEE dynamics over time, time derivatives of temperature and radiation become key components. The individual models for temperature and radiation are provided in the Appendix sections A.2 and A.3.

#### 3.1.1 Nighttime estimate based on the measurements

At night, GPP is assumed to be zero (when global radiation $R_g < 20 W.m^{-2}$). In this context, the measured NEE is essentially $R_{eco,t}$, which follows the temperature dependence of the Arrhenius-type Lloyd & Taylor (1994) :

$$R_{eco,t} = r_{night} \, exp\left( E_0 \left( \frac{1}{T_{ref} - T_0} - \frac{1}{T_{air,t} - T_0} \right) \right) \tag{2}$$

where $r_{night}$, in µmol C m$^{-2}$s$^{-1}$, is the base respiration at the reference temperature $T_{ref} = 15°C$, $E_0$, in $°C$, is the temperature sensitivity that is fixed for the whole year, $T_{air,t}$ is the air temperature and $T_0$ is the temperature constant and fixed as $-46.02°C$ Lasslop et al. (2010). For consistency of the model, $r_{night}$ is generally updated every 5 days using estimations based on 15-day windows of historical data Reichstein et al. (2005).

#### 3.1.2 Daytime estimate including temperature sensitivity respiration

During the day, GPP is assumed to be non-zero and therefore, the two components of NEE are defined as:

$$R_{eco,t} = r_{day} \, exp\left( E_0 \left( \frac{1}{T_{ref} - T_0} - \frac{1}{T_{air,t} - T_0} \right) \right), GPP_t = \frac{\alpha\beta R_{g,t}}{\alpha R_{g,t} + \beta} \tag{3}$$

Plant respiration is approximately 25% higher during the day compared to the night Jones et al. (2024), and thus the base respiration value is computed separately for daytime as $r_{day}$ which is updated every 5 days using estimations based on 15-day windows of historical data. As soil respiration is a very large component of $R_{eco,t}$ and continue from night to day, the temperature sensitivity of the respiration $E_0$ might not change. Thus, $E_0$ is estimated from the nighttime model and extrapolated to the daytime ecosystem respiration model for consistency with the data. $R_{g,t}$, in W.m$^{-2}$, is the global

radiation. $\alpha$, in µmol C J$^{-1}$, is the canopy light utilisation efficiency and $\beta^1$, in µmol C m$^{-2}$s$^{-1}$, is the maximum $CO_2$ uptake rate of the canopy at light saturation, updated over several weeks.

## 3.2 Net Ecosystem Exchange dynamics as a Stochastic Differential Equation

The NEE dynamics measured from flux tower are modeled as a Stochastic Differential Equation described by a Wiener process White & Luo (2008); Weng (2011), where the drift (deterministic part of NEE dynamics) is determined by the NEE models (section 3.1) and the diffusion coefficient (representing noise in NEE measurements) is calculated as the standard deviation of the distribution of the error between the measurement data and the NEE models. This noise is proved to be Gaussian (see Appendix A.4). Equation 4 describe the formulation of the SDE based on these two components. NEE evolves over time, fluctuating around the drift $\mu_t$ with a noise of the measurement distribution $\sigma_t dW_t$ (assumed to be Gaussian). The computation of the drift $\mu_t$ is derived from the decomposition of NEE, equation 1 and is given by Equation 5. Further details on the drift computation and noise assumption are provided in Appendix sections A.4 and A.5.

$$d\,\mathrm{NEE}_t = \mu_t dt + \sigma_t dW_t \tag{4}$$

with the drift

$$\mu_t = \frac{d\mathrm{NEE}_t}{dt} = \frac{d}{d\mathrm{T}_{\mathrm{air}}}\mathrm{R}_{\mathrm{eco},t}(\mathrm{T}_{\mathrm{air},t})\frac{d\mathrm{T}_{\mathrm{air},t}}{dt} - \frac{d}{d\mathrm{R}_{\mathrm{g}}}\mathrm{GPP}(\mathrm{R}_{\mathrm{g},t})\frac{d\mathrm{R}_{\mathrm{g},t}}{dt}$$

$$\frac{d}{d\mathrm{T}_{\mathrm{air}}}\mathrm{R}_{\mathrm{eco},t}(\mathrm{T}_{\mathrm{air},t}) = \frac{E_0}{(\mathrm{T}_{\mathrm{air},t} - T_0)^2}\mathrm{R}_{\mathrm{eco},t}, \frac{d\mathrm{T}_{\mathrm{air},t}}{dt} = \pi\frac{\Delta\mathrm{T}_{\mathrm{air},t}}{t_{day}}sin(2\pi\frac{t - t_{T_{max}}}{t_{day}})$$

$$\frac{d}{d\mathrm{R}_{\mathrm{g}}}\mathrm{GPP}_t(\mathrm{R}_{\mathrm{g},t}) = \frac{\alpha\beta^2}{(\alpha\mathrm{R}_{\mathrm{g},t} + \beta)^2}, \frac{d\mathrm{R}_{\mathrm{g},t}}{dt} = \frac{R_{lw,0}^{down}}{T_{air,0}}\frac{d\mathrm{T}_{\mathrm{air},t}}{dt} + (R_{sw}^{0,down} + R_{diff}^0)\frac{d}{dt}R_{norm,t}^{\odot} \tag{5}$$

The diffusion coefficient is constant and calculated separately for nighttime and daytime using the flux data associated with both times.

For the notation of the PIAE, we will denote the SDE 4 as

$$\mathrm{f}_t(\omega) = \mathcal{N}_t[\mathrm{T}_{\mathrm{air},t}(\omega), \mathrm{R}_{\mathrm{g},t}(\omega), k_t(\omega)], \; d\,\mathrm{NEE}_t = \mathrm{f}_t(\omega) + \varepsilon_t(\omega), \omega \in \Omega \tag{6}$$

where $\mathrm{NEE}_t$ is the solution of SDE, $k_t$ refers to the parameters of the day and night time models ($r_{night/day}$, $E_0$, $\alpha$ and $\beta$), $\mathrm{f}_t$ is the drift (also known as forcing term), $\varepsilon_t$ the noise and $\omega$ is a realisation in the probability space of the experiments $\Omega$.

## 4 Physics Informed Autoencoder for Net Ecosystem Exchange prediction

We propose a Physics-Informed Auto-Encoder (PIAE) to address gaps in NEE measurements, utilizing the SDE defined as a Wiener process (see equation 4) Zhong & Meidani (2023); White & Luo (2008). To fill these gaps, we estimate the NEE model parameters from measurements and then we solve the forward problem to estimate NEE where gaps exist White & Luo (2008) by integrating the SDE defined in Section 3.2 into the PIAE architecture.

Our PIAE architecture, inspired by Zhong & Meidani (2023), includes an Encoder module that compresses input variables into a latent space, feeding the latent vector to six decoders to reconstruct variables used as SDE components, NEE at the current time instance $t$ and the noise term. The predicted variables and the noise term are fed to the SDE ($\mathcal{N}_t$) to compute the change in NEE (drift term) $\frac{d\mathrm{NEE}_t}{dt}$. This is then added to the reconstructed NEE to forecast the NEE for time instance $t+1$.

The loss term is comprised of a weighted sum of two cost functions. Mean Square Error (MSE) is used to fit data point-wise to target $\mathrm{NEE}_{t+1}$, the reconstructed $\mathrm{NEE}_t$ and the SDE components. Maximum Mean Discrepancy (MMD) with Gaussian kernels $\{\ker_i\}_{i=1}^M$ is used to fit the target distribution and align the predicted noise term to the distribution of the target error distribution between

---

[1]vapor pressure deficit (VPD) limitation can also be taken into account Lasslop et al. (2010)

measurements and the NEE model (from section 3.1). The encoder and decoders are fully connected feed-forward neural network layers. The architecture is detailed in Figure 1 and Algorithm 1. To streamline predictions, we use temperature and radiation measurements from flux data as ground truth, bypassing the need to estimate individual components of the temperature and radiation models (see Sections A.2 and A.3 in the Appendix section).

## 4.1 Drift term $f_t$ and Estimating $\frac{d}{dt}T_{air,t}$, $\frac{d}{dt}R_{g,t}$

We describe our SDE as a stochastic process as drift term $f_t$ defined as:

$$f_t = \frac{dNEE_t}{dt}. \tag{7}$$

We calculate the ground-truth values for $f_t$ from the measurement data using the right-side first order approximation of the derivative:

$$\frac{dNEE_t}{dt} \approx \frac{NEE_{t+1} - NEE_t}{\Delta t} \tag{8}$$

Here, $\Delta t$ is 30min since the flux data measurements were recorded at 30 minutes intervals. As illustrated in Figure 1, we predict the values of the drift term $\tilde{f}_t$ based on the SDE 6 of the predicted $\widetilde{NEE}_t$ as a function of the predicted parameters $\tilde{k}_t$ and measurements for $T_{air,t}$ and $R_{g,t}$ as follows:

$$\frac{d\widetilde{NEE}_t}{dt} = \mathcal{N}_t(T_{air,t}, R_{g,t}, \tilde{k}_t) = \tilde{f}_t \tag{9}$$

The same method is applied to get ground-truth values of $\frac{d}{dt}T_{air,t}$ and $\frac{d}{dt}R_{g,t}$ which are predicted by the decoders in the architecture (discussed in Section 4.4).

## 4.2 Model Inputs and Target Variables

We define a set of input variables $X_t$ which contains the meteorological variables from table 1. We also define a set of ground-truth variables to optimize the model outputs against. This is described as $S_t = \{NEE_t, dT_{air}, dR_g, k_t, f_t, \varepsilon_t\}$. To reiterate, $k_t$ is defined as the set of NEE model parameters $k = (E_0, r_{night/day}, \alpha, \beta)$ based on day or night time model of NEE being considered. Here $\varepsilon_t$ is the noise term based on the standard deviation of the error in NEE between the measurements and the model. The use of the noise enables to consider the sensitivity regarding the initialization of the SDE.

## 4.3 Encoder

As described in the figure 1, the encoder $\mathcal{E}_\phi$ maps the input variables in $X_t$ and estimated parameters $k_t$ to the latent variable $z$, such that:

$$z^{(j)} = \mathcal{E}_\phi(X_t(\omega^{(j)}), k_t) \tag{10}$$

where $\omega^{(j)}$ represents the implicit realisation of the random event producing the measurements. This compresses the meteorological measurements and associated parameters to a latent space.

## 4.4 Decoders

As seen in the figure 1, six independent decoders denoted $\tilde{k}_{t,\theta}(z)$, $\widetilde{NEE}_{t,\theta}(z)$, $\tilde{f}_{t,\theta}(z)$, $\widetilde{\frac{d}{dt}T}_{air,t,\theta}(z)$, $\widetilde{\frac{d}{dt}R}_{g,t,\theta}(z)$ and $\widetilde{\varepsilon}_t(z)$ have been implemented to approximate the components of the stochastic process in $S_t$ respectively by constructing a mapping from the latent variable $z$ to the input space. Then, Inspired by the Physics-Informed Neural Networks for deterministic differential equations Raissi et al. (2017a;b), we incorporate the governing differential equation into the framework by applying the differential operators in $\mathcal{N}_t$ on the outputs of the decoders, to obtain an approximation of the $f$ in the governing SDE such that:

$$\tilde{f}_{t,\theta}(z) = \mathcal{N}_t[\widetilde{T_{air,t,\theta}}(z), \widetilde{R_{g,t,\theta}}(z), \tilde{k}_{t,\theta}(z)] \tag{11}$$

The decoder for the noise term $\widetilde{\varepsilon}_t(z)$ predicts the mean and log variance of the noise, which are then used to sample a noise value using the reparameterization trick as done in Variational Auto Encoders. Differentiation in $\mathcal{N}_t$ is done by the automatic differentiation technique Paszke et al. (2017), using the graph structure to compute gradients and allowing the PIAE model to learn during training without manual gradient computation. These physics-informed estimates $\tilde{f}_{t,\theta}(z)$ together with the approximated response $\widetilde{T}_{\text{air},t,\theta}(z), \widetilde{R}_{\text{g},t,\theta}(z)$ and the day and nighttime model parameters $\tilde{k}_{t,\theta}(z)$ constitute $N$ reconstructed snapshots i.e. $\{\tilde{S}_t(z^{(j)})\}_{j=1}^N$, described in the equation 12, where $\{\widetilde{\text{NEE}}_t(z^{(j)}), \tilde{K}_t(z^{(j)}), \widetilde{T}_{\text{air},t}(z^{(j)}), \widetilde{R}_{\text{g},t}(z^{(j)}), \tilde{f}_t(z^{(j)}), \widetilde{\varepsilon}_t(z^{(j)})\}_{j=1}^N$ are the reconstructed snapshots associated to the decoders mentioned above.

$$\{\tilde{S}_t(z^{(j)})\}_{j=1}^N = \{\widetilde{\text{NEE}}_t(z^{(j)}), \tilde{K}_t(z^{(j)}), \widetilde{T}_{\text{air},t}(z^{(j)}), \widetilde{R}_{\text{g},t}(z^{(j)}), \tilde{f}_t(z^{(j)}), \widetilde{\varepsilon}_t(z^{(j)})\}_{j=1}^N,$$
$$\tilde{K}_t(z^{(j)}) = \tilde{k}_{t,\theta}(z^{(j)}), \widetilde{T}_{\text{air},t}(z^{(j)}) = \widetilde{T}_{\text{air},t,\theta}(z^{(j)}), \widetilde{R}_{\text{g},t}(z^{(j)}) = \widetilde{R}_{\text{g},t,\theta}(z^{(j)}), \tag{12}$$
$$\tilde{f}_t(z^{(j)}) = \tilde{f}_{t,\theta}(z^{(j)}), \widetilde{\text{NEE}}_t(z^{(j)}) = \widetilde{\text{NEE}}_{t,\theta}(z^{(j)})$$

Consequentially, the forecast $\text{NEE}_{t+1}$ is calculated within the PIAE as:

$$\widetilde{\text{NEE}}_{t+1} = \widetilde{\text{NEE}}_{t,\theta}(z) + \tilde{f}_{t,\theta}(z) + \widetilde{\varepsilon}_{t,\theta}(z) \tag{13}$$

### 4.5 LOSS FUNCTION

The loss term is comprised of a weighted sum of two cost functions. Mean Square Error (MSE) is used to fit data point-wise to target $\text{NEE}_{t+1}$, the reconstructed variables $\text{NEE}_t, d\text{T}_{\text{air}}, d\text{R}_{\text{g}}, k_t$ and the drift term $f_t$. For explanation purposes, assume these variables form the ground-truth set $\mathcal{D}_1$ and prediction set $\widetilde{\mathcal{D}_1}$ as:

$$\mathcal{D}_1(\omega) = \{\text{NEE}_t(\omega), d\text{T}_{\text{air},t}(\omega), d\text{R}_{\text{g},t}(\omega), k_t(\omega), f_t(\omega)\}$$
$$\widetilde{\mathcal{D}_1}(z) = \{\widetilde{\text{NEE}}_t(z), \widetilde{T}_{\text{air},t}(z), \widetilde{R}_{\text{g},t}(z), \tilde{K}_t(z), \tilde{f}_t(z)\} \tag{14}$$

Maximum Mean Discrepancy (MMD) with Gaussian kernels $\{\text{ker}_i\}_{i=1}^M$ is used to fit the target distribution of $\text{NEE}_{t+1}$ and align the predicted noise term to the distribution of the target error distribution. Here, we can assume a ground-truth set $\mathcal{D}_2$ and prediction set $\widetilde{\mathcal{D}_2}$ as:

$$\mathcal{D}_2(\omega) = \{\text{NEE}_{t+1}(\omega), \varepsilon_t(\omega)\}, \ \widetilde{\mathcal{D}_2}(z) = \{\widetilde{\text{NEE}}_{t+1}(z), \widetilde{\varepsilon}_t(z)\} \tag{15}$$

The final cost function is computed using the weighted sum of MSE and MMD loss terms between the measurements $\{\mathcal{D}_1(\omega^{(j)})\}_{j=1}^N, \{\mathcal{D}_2(\omega^{(j)})\}_{j=1}^N$ and predicted samples $\{\widetilde{\mathcal{D}_1}(z^{(j)})\}_{j=1}^N, \{\widetilde{\mathcal{D}_2}(z^{(j)})\}_{j=1}^N$. Thus the given loss function is

$$\text{Loss}_{\text{mse}} = \text{MSE}(\widetilde{\mathcal{D}_1}(z), \mathcal{D}_1(\omega)), \text{Loss}_{\text{mmd}} = \sum_{i=1}^M \text{MMD}_{\text{ker}_i, \text{NEE}}(\widetilde{\mathcal{D}_2}(z), \mathcal{D}_2(\omega))$$
$$\text{MMD}_s(P, Q) = \mathbb{E}_{x,x'}[s(x, x')] + \mathbb{E}_{y,y'}[s(y, y')] - 2\mathbb{E}_{x,y}[s(x, y)] \tag{16}$$
$$\text{Loss} = w_{MSE} \, \text{Loss}_{\text{mse}} + w_{MMD} \, \text{Loss}_{\text{mmd}}$$

where for $x, x'$ in a data space $\boldsymbol{S}$ with probability distribution $P$ and $y, y'$ in the output space $\tilde{\boldsymbol{S}}$ with probability distribution $Q$. And $w_{MSE}, w_{MMD}$ the weights associated to the losses.

After training, the decoders $\tilde{k}_{t,\theta}(z)$, $\widetilde{\text{NEE}}_{t,\theta}(z)$, $\tilde{f}_{t,\theta}(z)$, $\frac{d}{dt}\widetilde{T}_{\text{air},t,\theta}(z)$ and $\frac{d}{dt}\widetilde{R}_{\text{g},t,\theta}(z), \widetilde{\varepsilon_{t,\theta}(z)}$ are equipped to approximate accurate values of the stochastic process components $k_t(\omega)$, $\text{NEE}_t(\omega)$, $\frac{d}{dt}\text{T}_{\text{air},t}(\omega)$, $\frac{d}{dt}\text{R}_{\text{g},t}(\omega)$, $\varepsilon_t(\omega)$. The loss terms on $\text{NEE}_{t+1}$ ensures that the forecasting operation in Equation 13 is an explicit part of the PIAE architecture and thereby gradient calculations.

## 5 EXPERIMENTS

We run our experiments for the night time flux data and corresponding model of NEE, which are incorporated into the PIAE architecture according to the Equations 5 and 6. The experiments on the day-time data and model can be found in Appendix section A.7.

---

**Algorithm 1** PIAE for SDE algorithm Zhong & Meidani (2023)

**Initialisation**:
Set the number of training steps $n_t$,
batch size $N$,
Adam hyperparameters $\alpha, \beta_1, \beta_2$,
Initial parameters for the encoder and the decoder $\phi, \theta_k, \theta_{\text{NEE}}, \theta_{\frac{d}{dt}\text{T}_{\text{air},t}}$ and $\theta_{\frac{d}{dt}\text{R}_{\text{g},t}}$,
$\text{ker}_i$ kernels of MMD estimators

  **for** $i = 1, \cdots, n_t$ **do**
    Sample $N$ snapshots $\{\text{S}(\omega^{(j)})\}_{j=1}^N$.
    **for** $j = 1, \cdots, N$ **do**
      $z^{(j)} = \mathcal{E}(\text{S}(\omega^{(j)}, k_t))$,
      $\widetilde{\text{NEE}}(z^{(j)}), \tilde{K}(z^{(j)}), \widetilde{\text{T}}_{\text{air}}(z^{(j)}), \widetilde{\text{R}}_{\text{g}}(z^{(j)}), \tilde{\text{f}}(z^{(j)}), \widetilde{\varepsilon}(z^{(j)})$
      $= \widetilde{\text{nee}}(z^{(j)}), \tilde{k}(z^{(j)}), \widetilde{\text{T}}_{\text{air},t},(z^{(j)}), \widetilde{\text{R}}_{\text{g},t}(z^{(j)}), \tilde{\text{f}}_t(z^{(j)}), \widetilde{\varepsilon}_t(z^{(j)})$
      $\tilde{\text{S}}(\omega^{(j)})$
      $= [\tilde{K}_\theta(z^{(j)}), \widetilde{\text{NEE}}(z^{(j)}), \tilde{\text{f}}(z^{(j)}), \tilde{R}_g(z^{(j)}), \tilde{T}_{air}(z^{(j)}), \widetilde{\varepsilon}(z^{(j)})]$,
    **end for**
    Loss
    $= w_{\text{MSE}}\text{MSE}(\{\text{S}(\omega^{(j)}\}_{j=1}^N, \{\tilde{\text{S}}(\omega^{(j)}\}_{j=1}^N)$
    $+ w_{\text{MMD}} \sum_i^N \text{MMD}_{\text{ker}_i,\text{NEE}}(\{\text{S}(\omega^{(j)}\}_{j=1}^N, \{\tilde{\text{S}}(\omega^{(j)}\}_{j=1}^N)$
    $\theta, \phi \leftarrow \text{Adam}(\nabla Loss, \theta, \phi, \alpha, \beta_1, \beta_2)$
  **end for**

---

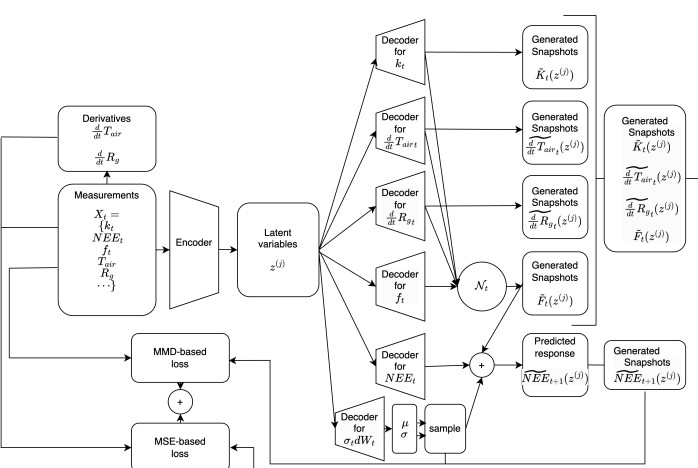

Figure 1: Architecture of the PIAE for SDE in order to fill the gaps in the NEE measured by the flux tower.

We compare our PIAE approach to three other conventional methods from the literature: RFR (Random Forest Robust) (current state of the art) Moffat et al. (2007), XGBoost (based on the same configuration as RFR) and a basic Autoencoder. The computational costs of PAIE, RFR and XG-Boost were similar computational costs on our machine, a laptop. The Autoencoder (AE) model comprises of an Encoder architecture similar to the Encoder in the PIAE model. There are two Decoders predicting parameters in $k$ ($E_0, r, \alpha, \beta$) and the NEE values at next timestamp, similar in number of parameters and layers as in PIAE model. The loss function is kept as MSE to align reconstructed $\tilde{k}_t$ and the forecasted $\widetilde{\text{NEE}}_{t+1}$ since the AE has no stochastic component. We also compare a version of our PIAE model where the decoder for $\widetilde{\text{NEE}}_t(z)$ is replaced by $\widetilde{\text{NEE}}_{t+1}(z)$ and where the noise term $\widetilde{\varepsilon}_t(z)$ is added to this value to predict the forecast value directly. Here, $\tilde{f}_t(z)$ is not directly used in the forecast and therefore the drift component $\mathcal{N}_t$ only acts as a regularization during training. We will label this as PIAE-RegOnly in our experiments herewith.

## 5.1 PARAMETER ESTIMATION $E_0, r_{night/day}, \alpha, \beta$

The flux tower data does not provide ground-truth values for the day and nighttime model parameters $E_0, r_{night/day}, \alpha, \beta$ introduced in the models section 3.1. Therefore, we use REddyProc partitioning algorithm to estimate from the flux tower measurements Wutzler et al. (2018). REddyProc is a non-linear regression method that estimates $E_0, r_{night/day}, \alpha, \beta$ based on the methodology of Reichstein et al. (2005) for the nighttime, and of Lasslop et al. (2010) for the daytime. In the work from Lasslop et al., we have existing prior knowledge on the range of $E_0$ the temperature sensitivity ($E_0 \in [50, 400]$), $r_{night/day}$ the base respiration at reference temperature $T_{ref}$ ($r_{day}, r_{night} > 0$), $\alpha$ the canopy light utilisation efficiency ($\alpha \in [0, 0.22)$) and $\beta$ the maximum $CO_2$ uptake ($\beta \in [0, 250)$) Lasslop et al. (2010).

We use two different methods for calculating the parameters for nighttime and daytime models. As explained in Section 3.1, the GPP value at night time is assumed to be zero. Therefore we can assume NEE to be calculated directly from $R_{eco}$ (see Equation 2). As such, for nighttime parameters $E_0$ and $r_{night/day}$, we follow a method based on the flux partitioning described in Reichstein et al. (2005): we divide the nighttime data into groups of data points representing each night (for points with radiation values greater than 10 W.m$^{-2}$). For each group, we estimate values for $E_0$ and $r_{night/day}$ by applying the Lloyd-and-Taylor model Lloyd & Taylor (1994) by fitting to the scatter of NEE and $T_{air,t}$ using non-linear regression.

For the daytime model, because of the introduction of GPP in the equation of NEE (see Equations 1 and 3), we calculate $E_0, r, \alpha, \beta$. We follow a method based on the flux partitioning defined by Lasslop et. al Lasslop et al. (2010). Here, for each daytime data group, we use the same $E_0$ values estimated from the nighttime data of the respective day (previously calculated). We first estimate values for $\alpha$ and $\beta$ by fitting them to the scatter of GPP and $R_g$ using non-linear regression. Consequentially, with the estimated values for $E_0, \alpha, \beta$ and NEE values for each daytime data group, we use Equation 3 to calculate $r_{night/day}$ for each data point in the daytime data group.

## 5.2 TRAINING DATA CONFIGURATION

For both day and night time modes, we divide the flux measurements (and corresponding estimated parameters) into training and testing datasets based on yearly data. The training dataset comprises data from six years (2012 to 2017) with approximately 21000 data points while the testing dataset comprises data from two years (2018, 2019) with approximately 5300 data points.

Table 2: Results for NEE prediction on Night time data and model experiments. The metrics MMD, Wasstn (Wassertein Distance), KL (Kullback Leibler Divergence), and MAE (Mean Absolute Error) are expressed as the lower the better. R2 (score) is expressed as higher the better.

| Approach | MMD | Wasstn | KL | MAE | R2 |
|---|---|---|---|---|---|
| **PIAE** | **$0.025 \pm 0.003$** | **$0.140 \pm 0.018$** | **$0.069 \pm 0.012$** | **$0.851 \pm 0.01$** | **$0.73 \pm 0.002$** |
| **PIAE-RegOnly** | **$0.035 \pm 0.002$** | **$0.194 \pm 0.01$** | **$0.110 \pm 0.015$** | **$0.867 \pm 0.009$** | **$0.74 \pm 0.004$** |
| AE | $0.047 \pm 0.002$ | $0.185 \pm 0.008$ | $0.210 \pm 0.013$ | $0.84 \pm 0.001$ | $0.74 \pm 0.001$ |
| RF | $0.055 \pm 0.001$ | $0.237 \pm 0.001$ | $0.242 \pm 0.006$ | $0.901 \pm 0.002$ | $0.721 \pm 0.002$ |
| XGB | $0.052 \pm 0$ | $0.202 \pm 10^{-17}$ | $0.214 \pm 10^{-17}$ | $0.988 \pm 10^{-17}$ | $0.658 \pm 0$ |

## 5.3 RESULTS

We evaluated the methods using three distribution-based metrics: Mean Maximum Discrepancy (MMD), Wasserstein Distance, and Kullback-Leibler Divergence (KL), to assess how well each technique captures the distribution of the target variables. Additionally, we measured performance using mean absolute error (MAE) and R2 score to evaluate the fit to target variables.

Table 2 summarizes the results. Our PIAE method outperforms state-of-the-art tree-based methods (RF and XGB) on both distribution metrics and MAE/R2 scores for nighttime data. At best, the PIAE reports 34.6% lower KL score than RF and 45% lower an MMD score. The vanilla Autoencoder

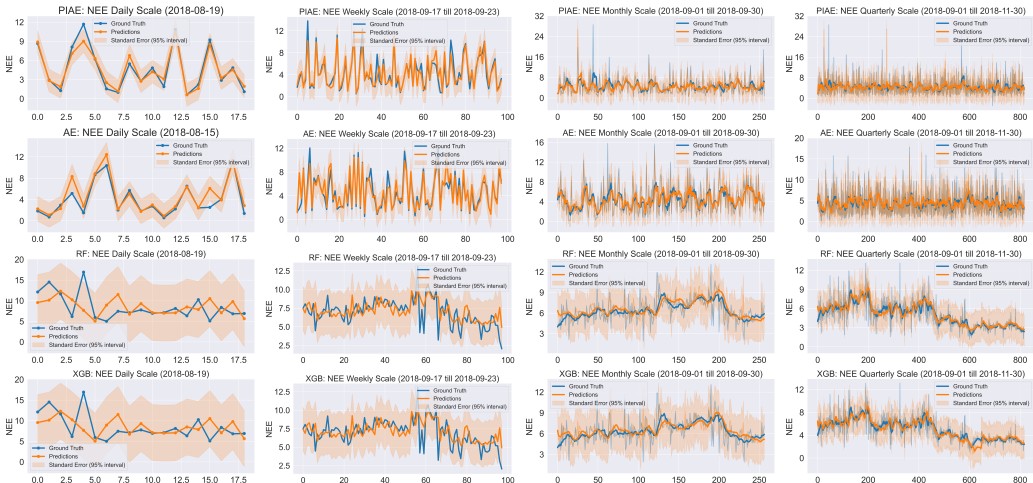

Figure 2: Night time model results on test data across different time scales for each approach in the experimentation. Row 1 represents results based on the PIAE model, Row 2 represents results from the AE model, Row 3 represents results from the RF model, and Row 4 represents results from the XGB model respectively. The sequences illustrated in the graphs are randomly sampled from the test dataset and are kept consistent for each approach for fair validation. The actual timestamps of the sequences are mentioned at the top of each graph.

also surpasses tree-based methods on both data fit and distribution based metrics. The physics-based decoder architecture and loss function in PIAE allow it to exceed the vanilla auto-encoder, especially on the distribution-based scores, with the incorporation of the associated SDE.

Figure 2 illustrates results across five time scales—daily, weekly, monthly, and quarterly—on the nighttime test dataset. Randomly sampled sequences, consistent across methods for fair validation, show that PIAE consistently captures NEE trends better than all other methods, highlighting the benefit of the incorporated SDE. The AE approach also performs well compared to RF and XGB, though it lags behind PIAE.

Consistent with Moffat et al. (2007), our experiments confirm that Random Forest performs better for daytime NEE modeling. For the daytime test dataset, PIAE showed similar performance to AE, RF, and XGB across all metrics and can be found in Table 4 in the Appendix section A.7.

PIAE and AE models have an advantage over RF and XGB in predicting parameters of the stochastic differential equation using dedicated decoders. Tables 5 and 6, in the appendix, compare the predictions for parameters $E_0, r_{night/day}, \alpha, \beta$ for nighttime and daytime. Both models achieve similar accuracy for daytime parameters, with AE slightly better at modeling nighttime parameters. It is important to note that these parameters were estimated based on a non-linear regression of the scatter of NEE, $T_{air,t}$ and $R_g$ values and were intended to guide the learning of the next NEE in the decoder. Thus, PIAE and AE give good results with a similar MAE and a similar high R2. To sum up, PIAE and AE offers a close estimation of the parameters of NEE dynamics modeled as a stochastic differential equation.

## 6 DISCUSSION AND CONCLUSION

In this study, we introduced a Physics-Informed Autoencoder (PIAE) to address the forward problem of Net Ecosystem Exchange (NEE) gap-filling, utilizing a Stochastic Differential Equation (SDE) to enhance the quality of $CO_2$ measurements from flux towers at the agricultural field scale. This approach not only improves the data quality for training Net Ecosystem Exchange forecasting methods but also integrates deterministic models for nighttime and daytime Net Ecosystem Exchange alongside stochastic components, such as Gaussian noise.

In Section 3, we outlined the Net Ecosystem Exchange models that account for both deterministic phenomena and stochastic uncertainties in the measurements. Section 4 detailed our Physics-

Informed Autoencoder, which effectively addresses the forward problem by estimating model parameters. In Section 5, we demonstrated that Physics-Informed Autoencoder outperforms state-of-the-art methods by approximately 22% in R2 score and 52% in MMD score for nighttime Net Ecosystem Exchange gaps and captures trends across daily to quarterly scales more effectively.

Our method's effectiveness is further validated by comparing it with a standard Autoencoder (AE), particularly for nighttime data, where Physics-Informed Autoencoder significantly outperforms Autoencoder due to the integration of the Stochastic Differential Equation. Additionally, Physics-Informed Autoencoder performs gap-filling in a duration-agnostic manner, similar to the Random Forest Robust (RFR) method, but with the added advantage of incorporating physical laws through Stochastic Differential Equation. Furthermore, Physics-Informed Autoencoder offers forecasting capabilities by predicting Net Ecosystem Exchange at the next time instance enhancing its utility beyond gap-filling.

In conclusion, the use of Physics-Informed Autoencoder for Net Ecosystem Exchange stochastic dynamics has successfully filled gaps ranging from half-hourly to yearly in Net Ecosystem Exchange measurements from the flux tower in East Anglia fields, with satisfactory uncertainty levels for both day and night. This method is ready for deployment and will serve as a robust example for Digital Twin projects, such as AI4NetZero, aimed at climate change monitoring.

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
