## A APPENDIX

### A.1 FLUX TOWER PICTURE

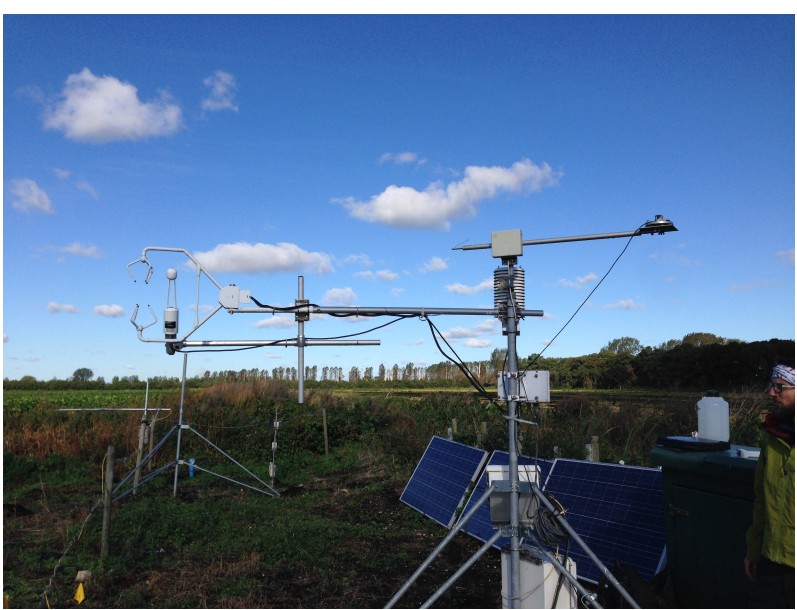

Figure 3: Flux tower in East Anglia, UK, composed of a sonic anemometer, an infrared analyser on the arm of the tower at the left of the picture

### A.2 TEMPERATURE MODEL

A key component in expressing the dynamics of NEE as a function of time White & Luo (2008); Weng (2011) is the differential daily temperature model, which is described in the JULES, the Joint UK Landscape Simulator Williams & Clark (2014), as the equation:

$$\mathrm{T}_{\mathrm{air},t} = T_{air,r} + \frac{\Delta \mathrm{T}_{\mathrm{air},t}}{2} cos(2\pi \frac{t - t_{T_{max}}}{t_{day}}) \tag{17}$$

where $T_{air,r}$ and $\Delta \mathrm{T}_{\mathrm{air},t}$ are the temperature and diurnal temperature ranges respectively, $t_{day}$ is the length of a day and $t_{T_{max}}$ is the time of day where the temperature is highest. We assume $t_{T_{max}}$ occurs 0.15 of the length of the day after local noon time as:

$$t_{T_{max}} = \frac{t_{up} + t_{down}}{2} + 0.15(t_{up} - t_{down}) \tag{18}$$

with $t_{up}$, $t_{down}$ are the local sunrise and the sunset times.

### A.3 RADIATION MODEL

Another key component in expressing the dynamics of NEE as a function of time is the differential radiation model which is described in JULES as being based on the downward longwave $R_{sw}^{down}$, shortwave $R_{sw}^{down}$ and the diffuse radiation $R_{diff}$. The downward longwave solar radiation is defined as:

$$R_{lw,t}^{down} = R_{lw,0}^{down}(4\frac{\mathrm{T}_{\mathrm{air},t}}{T_{air,r}} - 3) \tag{19}$$

where $R_{lw,0}^{down}$ is the downward longwave solar radiation. Furthermore, the downward shortwave solar radiation is defined as:

$$R_{sw,t}^{down} = R_{sw}^{0,down} R_{norm,t}^{\odot} \tag{20}$$

where $R^{\odot}_{norm,t}$ is the solar radiation normalisation factor depending on the time $t$ and $R^{0,down}_{sw}$ is the downward shortwave solar radiation. The diffuse radiation is thus defined as:

$$R_{diff,t} = R^{0}_{diff} R^{\odot}_{norm,t}. \tag{21}$$

The constants $R^{down}_{lw,0}, R^{0,down}_{sw}, R^{0}_{diff}$ are fixed based on geo-scientific studies. The normalisation solar radiation $R^{\odot}_{norm,t}$ comes from Huntingford et al. (2010).

All in all, we consider the global radiation as:

$$\mathrm{R}_{g,t} = R^{down}_{lw,t} + R^{down}_{sw,t} + R_{diff,t} \tag{22}$$

## A.4 Drift of the Stochastic Differential Equation

Here are the drift details of the Wiener process. Firstly, we use the equation 17

$$\frac{d\mathrm{T}_{air,t}}{dt} = \pi \frac{\Delta \mathrm{T}_{air,t}}{t_{day}} sin(2\pi \frac{t - t_{T_{max}}}{t_{day}}). \tag{23}$$

Moreover, we consider the Arrhenius-type law of ecosystem respiration in equations 2 and 3

$$\frac{d}{d\mathrm{T}_{air}} \mathrm{R}_{eco,t}(\mathrm{T}_{air,t}) = \frac{E_0}{(\mathrm{T}_{air,t} - T_0)^2} \mathrm{R}_{eco,t} \tag{24}$$

where $r_{night/day}$ is a generic base respiration at reference temperature $T_{ref}$.

Then, we use the global radiation equation 22

$$\frac{d\mathrm{R}_{g,t}}{dt} = \frac{d}{dt}(R^{down}_{lw,t} + R^{down}_{sw,t} + R_{diff,t}) \tag{25}$$

Where each component of the sum is computed from the equations 19, 20 and 21, we deduce

$$\begin{aligned}
\frac{d}{dt}R^{down}_{lw,t} &= \frac{R^{down}_{lw,0}}{T_{air,0}} \frac{d\mathrm{T}_{air,t}}{dt} \\
\frac{d}{dt}R^{down}_{sw} &= R^{0,down}_{sw} \frac{d}{dt} R^{\odot}_{norm,t} \\
\frac{d}{dt}R_{diff,t} &= R^{0}_{diff} \frac{d}{dt} R^{\odot}_{norm,t}
\end{aligned} \tag{26}$$

where $\frac{d}{dt}R^{\odot}_{norm,t}$ is deduced from the IMOGEN routine `sunny` from JULES Huntingford et al. (2010).

Moreover, we deduce from the daytime equation model 3

$$\frac{d}{d\mathrm{R}_g} \mathrm{GPP}_t(\mathrm{R}_{g,t}) = \frac{\alpha\beta^2}{(\alpha\mathrm{R}_{g,t} + \beta)^2} \tag{27}$$

All things considered, we have the analytic expression for

$$\mu_t = \frac{d\mathrm{NEE}_t}{dt} = \frac{d}{d\mathrm{T}_{air}} \mathrm{R}_{eco,t}(\mathrm{T}_{air,t}) \frac{d\mathrm{T}_{air,t}}{dt} - \frac{d}{d\mathrm{R}_g} \mathrm{GPP}_t(\mathrm{R}_{g,t}) \frac{d\mathrm{R}_{g,t}}{dt} \tag{28}$$

## A.5 Diffusion coefficient, noise of measurements

We consider the diffusion coefficient for both nighttime and daytime $\sigma_{night}, \sigma_{day}$ that can be computed from the histograms of the error between the NEE values in flux data and the day/night Physics Models defined in section 3.1. We confirmed the Gaussian assumption using normality tests White & Luo (2008). These coefficients will be learned implicitly by the PIAE.

## A.6 PIAE and AE Neural Network Configurations

## A.7 Daytime Model Results

## A.8 Performance on predictions on nightime and daytime NEE parameters both PIAE and AE

```
PIAE_SDE(
  (encoder): Sequential(
    (0): Linear(in_features=18, out_features=16, bias=True)
    (1): ReLU()
    (2): Linear(in_features=16, out_features=16, bias=True)
    (3): ReLU()
    (4): Linear(in_features=16, out_features=32, bias=True)
  )
  (nee_decoder): Sequential(
    (0): Linear(in_features=32, out_features=16, bias=True)
    (1): ReLU()
    (2): Linear(in_features=16, out_features=16, bias=True)
    (3): ReLU()
    (4): Linear(in_features=16, out_features=1, bias=True)
  )
  (fc_mu): Sequential(
    (0): Linear(in_features=32, out_features=4, bias=True)
    (1): ReLU()
    (2): Linear(in_features=4, out_features=1, bias=True)
  )
  (fc_logvar): Sequential(
    (0): Linear(in_features=32, out_features=4, bias=True)
    (1): ReLU()
    (2): Linear(in_features=4, out_features=1, bias=True)
  )
  (temp_derivative_decoder): Sequential(
    (0): Linear(in_features=32, out_features=16, bias=True)
    (1): ReLU()
    (2): Linear(in_features=16, out_features=16, bias=True)
    (3): ReLU()
    (4): Linear(in_features=16, out_features=1, bias=True)
  )
  (k_decoder): Sequential(
    (0): Linear(in_features=32, out_features=16, bias=True)
    (1): LeakyReLU(negative_slope=0.01)
    (2): Linear(in_features=16, out_features=16, bias=True)
    (3): LeakyReLU(negative_slope=0.01)
    (4): Linear(in_features=16, out_features=2, bias=True)
    (5): LeakyReLU(negative_slope=0.01)
  )
)
```

Figure 4: Neural Network Configuration for PIAE Model

```
AE(
  (encoder): Sequential(
    (0): Linear(in_features=18, out_features=16, bias=True)
    (1): ReLU()
    (2): Linear(in_features=16, out_features=32, bias=True)
    (3): ReLU()
  )
  (nee_decoder): Sequential(
    (0): Linear(in_features=32, out_features=16, bias=True)
    (1): ReLU()
    (2): Linear(in_features=16, out_features=16, bias=True)
    (3): ReLU()
    (4): Linear(in_features=16, out_features=1, bias=True)
  )
  (k_decoder): Sequential(
    (0): Linear(in_features=32, out_features=16, bias=True)
    (1): ReLU()
    (2): Linear(in_features=16, out_features=16, bias=True)
    (3): ReLU()
    (4): Linear(in_features=16, out_features=2, bias=True)
  )
)
```

Figure 5: Neural Network Configuration for AE Model

Table 3: Tree Configuration using Python Sci-kit Learn Library for the RF and XGB models

|  | Tree Model | Number of Trees | Max Depth | Max Leaves |
|---|---|---|---|---|
| **RF** | 100 | None | None | None |
| **XGB** | 100 | None | None | None |

Table 4: Results for NEE prediction onDay time data and model experiments. The metrics MMD, Wasstn (Wassertein Distance), KL (Kullback Leibler Divergence), and MAE (Mean Absolute Error) are expressed as the lower the better. R2 (score) is expressed as higher the better.

| Approach | MMD | Wasstn | KL | MAE | R2 |
|---|---|---|---|---|---|
| PIAE | $0.027 \pm 0.004$ | $0.135 \pm 0.01$ | $0.042 \pm 0.003$ | $1.521 \pm 0.09$ | $0.868 \pm 0.03$ |
| AE | $0.026 \pm 0.003$ | $0.190 \pm 0.02$ | $0.071 \pm 0.005$ | $1.452 \pm 0.08$ | $0.877 \pm 0.03$ |
| RF | $0.050 \pm 0.005$ | $0.350 \pm 0.03$ | $0.149 \pm 0.010$ | $1.593 \pm 0.10$ | $0.857 \pm 0.04$ |
| XGB | $0.033 \pm 0.004$ | $0.234 \pm 0.02$ | $0.976 \pm 0.020$ | $1.532 \pm 0.09$ | $0.863 \pm 0.03$ |

Table 5: Performance on predictions of nighttime model parameters for both PIAE and AE models

|  | MMD | Wasstn | KL | MAE | R2 |
|---|---|---|---|---|---|
| $E_0$ (**PIAE**) | 0.172 | 3.388 | 1.237 | 5.629 | 0.941 |
| $E_0$ (**AE**) | 0.151 | 1.519 | 0.835 | 3.075 | 0.957 |
| $r_{night/day}$ (**PIAE**) | 0.124 | 0.304 | 0.99 | 0.437 | 0.942 |
| $r_{night/day}$ (**AE**) | 0.033 | 0.122 | 0.731 | 0.253 | 0.971 |

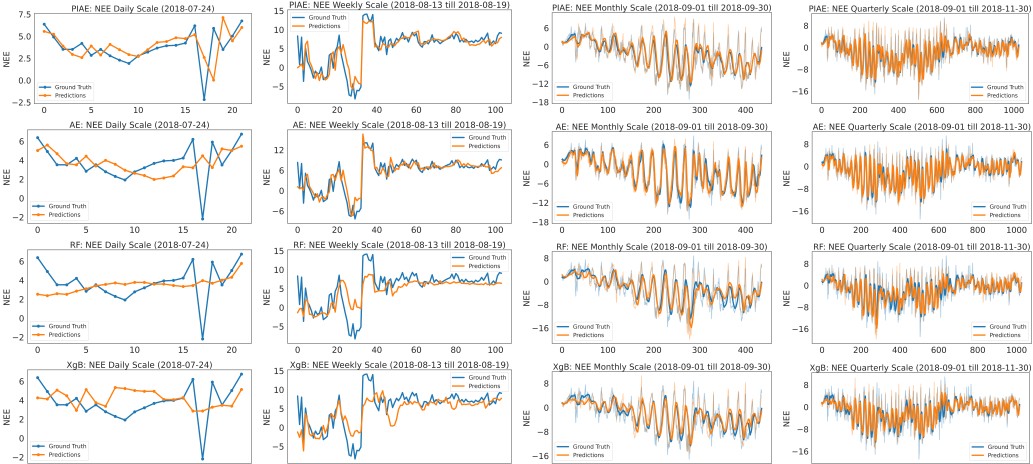

Figure 6: Day time model results on test data across different time scales for each approach in the experimentation. Row 1 represents results based on the PIAE model, Row 2 represents results from the AE model, Row 3 represents results from the RF model, and Row 4 represents results from the XGB model respectively. The sequences illustrated in the graphs are randomly sampled from the test dataset and are kept consistent for each approach for fair validation. The actual timestamps of the sequences are mentioned at the top of each graph.

Table 6: Performance on predictions of daytime model parameters for both PIAE and AE models

|  | MMD | Wasstn | KL | MAE | R2 |
|---|---|---|---|---|---|
| $E_0$ (**PIAE**) | 0.155 | 1.44 | 0.590 | 3.407 | 0.97 |
| $E_0$ (**AE**) | 0.118 | 1.145 | 0.465 | 2.236 | 0.982 |
| $r_{night/day}$ (**PIAE**) | 0.018 | 0.047 | 0.071 | 0.22 | 0.978 |
| $r_{night/day}$ (**AE**) | 0.037 | 0.091 | 0.067 | 0.247 | 0.977 |
| $\alpha$ (**PIAE**) | 0.0007 | 0.0079 | 1.139 | 0.0163 | 0.859 |
| $\alpha$ (**AE**) | 0.0196 | 0.0251 | 0.959 | 0.048 | -1.117 |
| $\beta$ (**PIAE**) | 0.285 | 3.502 | 0.594 | 6.182 | 0.984 |
| $\beta$ (**AE**) | 0.239 | 2.212 | 0.348 | 3.387 | 0.997 |