# OpenReview forum: "Physics-Informed Autoencoder for Enhancing Data Quality to Improve the Forecasting Reliability of Carbon Dioxide Emissions from Agricultural Fields"
_ICLR.cc/2025/Conference — Submitted to ICLR 2025_

### Official Review · Reviewer_oYun · 2024-10-30

**Soundness:** 3
**Presentation:** 2
**Contribution:** 1
**Rating:** 3
**Confidence:** 5

**Summary:**

In this study, the authors utilized physics-informed neural network framework (PINN) to develop an auto-encoder for the forecasting of carbon dioxide emission. First, the overall physical process is modeled by using a set of ordinary differential equations and PINN is used to train a neural network. The authors proposed a two-stage training method, that in the first stage the neural network is trained by minimizing the mean absolute error and, then, in the second phase, the maximum mean discrepancy score is minimized. It is shown that the proposed method outperforms some of the naive baselines.

**Strengths:**

It is shown that the proposed method outperformed some of the baseline methods that are used in the domain. It seems to suggest a potential of replacing the conventional machine learning models with the PINN models.

**Weaknesses:**

First of all, the study is mainly focused on the application of the widely used PINN to a physical process for a specific domain. It does not look like there are novel algorithms or problem setups that can be of interest to a broader machine learning community. I would like to suggest the authors to submit this manuscript to a more domain specific venue.

The paper is not very well written. It is unclear how the SDE formulation is treated in the modeling, how the SDE and model are used for uncertainty quantification, how the evaluations were made by using what variables as inputs and predict how long in the future, and so on. I assume that this is due to the page limitation. It would have been better if the authors had put all the domain specific modeling sections in the appendix and focused more  on the generic problem set up in the main body.

**Questions:**

The use of MMD seems a little bit odd. MMD is essentially a two-sample statistical test to identify of those samples are from the same probability distribution. We usually expect the two samples are from two independent realizations. But, based on the loss function, two samples are from the same realization, just one is the data and the other is a model prediction. If they are from the same realization ($\omega^j$ in author's notation), minimizing the distance would make more sense, like the first phase of the training. In the end of day, for two samples from the same realization, minimizing MMD corresponds to minimizing MSE. But, all the hyperparameters (like the RBF kernel) makes it much less straightforward.

---

> ### Author Response · Authors · 2024-11-25
> **General updates to the manuscript**
>
> Thank you for your comprehensive review of the manuscript. This has helped us significantly improve the content of the theoretical approach and experimentation. Here are the general updates we made to the manuscript:
>
> 1. Improved the Physics Informed Auto Encoder architecture to be consistent with the components of the SDE formalized in section 3.2 of the original manuscript. We have added another decoder that probabilistically predicts the noise term in the SDE i.e., $\sigma_t dW_t$ which is added to the decoder output for $NEE_{t+1}$ to generate the final value of $NEE_{t+1}$. This is done similar to how Variational Auto Encoder construct the probabilistic latent space with predicted values of $\mu$ and $log(\sigma^2)$. In our case, we sample a single value for noise term $\sigma_t dW_t$ using the noise decoder outputs of $\mu_{noise}$ and $log(\sigma_{noise}^2)$. According to White and Luo (2008) (referred to in line 168), for NEE modelling using SDEs, this noise term is same for both $dNEE_t$ and $NEE_t$, and is therefore also added to the drift $\mu_t dt$ in the SDE $\mathcal{N}_t$ in the PIAE architecture. This has led to a novel architecture for incorporating SDEs of this kind into Physics Informed Auto Encoder setups.
>
> 2. We analyzed the error distribution between the Physics Model defined in section 3.1 and the observed NEE values in the data. Based on the normality test, we show that this distribution is Gaussian and, therefore, can be used as ground truth for the noise in the data. The mean and standard deviation of this error distribution serve as the target distribution for our newly introduced noise decoder to converge towards.
>
> 3. Merged the two phases of the loss convergence to make them coherent based on the argument from Reviewer 4. We apply Maximum Mean Discrepancy (MMD) on the predicted stochastic NEE (with noise), while the Mean Squared Error (MSE) is used for all other decoder outputs (deterministic). The noise term is also aligned to the target error distribution (see point 2), using MMD. The training routine is now merged into a single phase with two weighted terms for both loss functions used.
>
> 4. Based on these improvements, we present significant improvements over original manuscript to all quoted metrics on the nighttime data. We include uncertainty analysis using standard errors for all quoted metrics (experimented over multiple runs of training and inference to cater for uncertainty resulting from model initialization and probabilistic predictions of noise term). For complete reproducibility, in the Appendix section, we add the complete experimentation, including all seed values used, hyperparameters, weight initialization technique and architecture specifications. The link to the Github repository containing the code is also included in the updated manuscript.
>
> 5. General improvements to the explanation (addressing reviewer remarks on SDE formalization, variables used etc.) and presentation of the concepts in the paper.
>
> We hope to share the updated manuscript before the stated deadline and look forward to your reviews.

---

> > ### Author Response · Authors · 2024-12-02
> > **Response after corrections, for Reviewer oYun**
> >
> > > First of all, the study is mainly focused on the application of the widely used PINN to a physical process for a specific domain. It does not look like there are novel algorithms or problem setups that can be of interest to a broader machine learning community. I would like to suggest the authors to submit this manuscript to a more domain specific venue.
> >
> > Now, we have made improvements to the approach which include several novel elements in terms of the approach.
> >
> > As seen in figure 1, we now introduced a stochastic sampler for the noise. The novelty now lies in the introduction of PIAE for SDE with a stochastic sampler. This approach can be extended to other applications where we can integrate SDEs of this form into autoencoders by predicting both the drift term and the noise term associated with the physical process being modelled. This also adds interpretability to the model. You can refer to the updated manuscript for the complete description.
> >
> > We hope this is interesting for the broader ICLR community now.
> >
> > > The paper is not very well written. It is unclear how the SDE formulation is treated in the modeling, how the SDE and model are used for uncertainty quantification, how the evaluations were made by using what variables as inputs and predict how long in the future, and so on. I assume that this is due to the page limitation. It would have been better if the authors had put all the domain specific modeling sections in the appendix and focused more on the generic problem set up in the main body.
> >
> > The SDE formulation and its use in the PIAE model are now consistent int the latest updates with the decoders in PIAE associated with the drift and noise term in the SDE in equation 4 and 5. We have also updated equation 6 to reflect its use and its coherence with the architecture in figure 1. With an explicit noise term in the decoder which is aligned, while training, to the error distribution between the physics model and the NEE measurements, uncertainty quantification is directly catered. This is described in section 4. in line 214-215.
> >
> > We show that the approach can provide accurate prediction till up to the quarterly scale assuming we have the input variables available. This assumption is also made by the current state-of-the-art Zhu et al. 2022 [1] in NEE gap-filling.
> >
> > > Questions : The use of MMD seems a little bit odd. MMD is essentially a two-sample statistical test to identify of those samples are from the same probability distribution. We usually expect the two samples are from two independent realizations. But, based on the loss function, two samples are from the same realization, just one is the data and the other is a model prediction. If they are from the same realization (
> >  in author's notation), minimizing the distance would make more sense, like the first phase of the training. In the end of day, for two samples from the same realization, minimizing MMD corresponds to minimizing MSE. But, all the hyperparameters (like the RBF kernel) makes it much less straightforward.
> >
> > This comment was very valuable and presented a valid argument. The update to the approach in the latest manuscript was based on this comment. We now have a single training phase with weighted sum of MMD and MSE loss functions. The MMD is only applied to the probabilistic outputs noise and $NEE_{t+1}$. We updated the calculation of $NEE_{t+1}$ in PIAE architecture with the introduction of a stochastic noise term, the use MMD to align this distribution to the target one is now theoretically valid, as seen in equation 13. MMD is also used to fit the target distribution and align the predicted noise term to the distribution of the target error distribution between measurements and the physics model. Please see the updates to introduction of section 4 and 4.4. and 4.5.
> >
> > **References**
> >
> > [1] Stable gap-filling for longer eddy covariance data gaps: A globally validated machine-learning approach for carbon dioxide, water, and energy fluxes, S. Zhu and R. Clement and J. McCalmont and C. Davies and T. Hill, Agricultural and Forest Meteorology, 2022, 314, 108777, 0168-1923, https://doi.org/10.1016/j.agrformet.2021.108777

---

> > > ### Comment · Reviewer_oYun · 2024-12-02
> > >
> > > I would like to thank the authors for the replies. Other than the adoption of MMD, it still follows the pretty much the same structure of PINN. The use of MMD is not fully justified. As the authors use a SDE with the Wiener process, the target variable has a normal distribution. Then, the standard log likelihood would be the optimal choice. In theory, MMD is flexible and capable of comparing any two arbitrary distributions, in practice, the use of the kernel and associated hyperparamters make it difficult to use for the training of neural networks. Moreover, in the UQ domain, it is usual to assume that all the variables, e.g. $T_{air}$, $k_t$, and $R_g$, are random variables and investigate the propagation of uncertainty through the governing equations, which makes the target variable $NEE$ non-Gaussian. However, in this study, it is simplified that only the target is a random variable and tried to learn it from the data. Then, it is hard to justify why MMD is used instead of the optimal and standard choice of negative log likelihood function.

---

> > > > ### Author Response · Authors · 2024-12-03
> > > > **Answer to the Official Comment by Reviewer oYun**
> > > >
> > > > Thank you for your quick answer and advice.
> > > >
> > > > The updated approach of PIAE embeds the Physics Equation directly into the neural network architecture – as opposed to a conventional PINN. Here $\mathcal{N_t}$ composed of the differential operators from Equation 5 are part of the neural architecture where gradients are backpropagated through during training since the final $NEE_{t+1}$ value is a combination of the output of $\mathcal{N_t}$ i.e., $dNEE_{t}$, the current NEE and the noise term. Also, conventional PINNs do not incorporate SDE’s in this form where we separately model the drift and noise terms which improves interpretability of the model.
> > > >
> > > > With regards to the argument on using loglikelihood is valid. In the future versions of this work, will definitely add a study using loglikelihood as the loss function and compare the results. Currently, the use of MMD may also be considered justified especially since we claim the distribution of target NEE in the flux measurements and the error distribution (between physics model and NEE measurements) is Gaussian. We have chosen Gaussian kernel as in Zhong and Meidani 2023 [1]. As stated in Zhong & Meidani 2023 [1] and Gretton et al. [2], the evaluation of MMD with various characteristic kernels has been shown to be consistent.
> > > >
> > > > **References**
> > > >
> > > > [1] PI-VAE: Physics-Informed Variational Auto-Encoder for stochastic differential equations, W. Zhong and H. Meidani , Computer Methods in Applied Mechanics and Engineering, Volume 403, Part A, 2023, https://doi.org/10.1016/j.cma.2022.115664.
> > > >
> > > > [2] A kernel two-sample set, Gretton A., Borgwardt K.M., Rasch M.J., Schölkopf B., Smola A., Journal of Machine Learning Research, 13, pp. 723 - 773, 2012, https://www.scopus.com/inward/record.uri?eid=2-s2.0-84859477054&partnerID=40&md5=f97ffe4fcf56556ac7c5f2822d03a841

---

### Official Review · Reviewer_mYoA · 2024-10-31

**Soundness:** 2
**Presentation:** 1
**Contribution:** 2
**Rating:** 5
**Confidence:** 2

**Summary:**

The paper addresses the problem of forecasting CO2 emission from agricultural fields based on measurement data. In particular, the problem of predicting missing data is addressed. The authors present a set of stochastic differential equations that govern the net ecosystem exchange (NEE) that are used in a physics-informed autoencoder for data imputation.

**Strengths:**

The proposed method seems a good fit to the application.

**Weaknesses:**

The paper was challenging to follow, primarily due to unclear notation and insufficient definitions of certain terms. Additionally, some design choices, such as the two-phase loss, are described but lack clear justification.

The main contribution appears to be a relatively straightforward application of an existing methodology to a specific domain. The novelty largely lies in application-specific details, which may not align closely with the primary interests of the ICLR community.

**Questions:**

Why is latent heat (L) excluded? Having a high correlation to the target variable would seem to be a good thing when the goal is to predict missing values?

The notation in equation 4-5 is difficult to read. Would it not be more clear to write this in terms of partial derivatives?

For improved readability, consider to use italics for variables and roman (upright) type for named functions, as subscripts in equations, and for units of measurement. Consider that multi-letter abbreviations can be confusing in equations: For example, it can be unclear if rb is a single variable or the product of r and b.

rb (night/day) is not defined in the main text as far as I can see. rb is mentioned in the text in the appendix but not in the mathematical derivations.

In equation 9, should is there not a difference between dt on the left and right hand side? On the left hand side, it seems to denote an infinitessimal element, and on the right side it is 30 minutes?

I am not sure how this approach is an autoencoder. As I understand the written description, the model predicts one timestep ahead with a latent encoding, and thus does forecasting rather than reconstruction. However, Figure 1 does seem to imply that the decoders predict for the same timestep.

Is there something wrong with the linebreaks in Algorithm 1, step 4?

What is the reason for the choice of the two loss phases?

I am not familiar with the literature on physics-informed autoencoders, but I would like to ask whether this paper introduces any technical contributions to the framework itself, or if the contribution is primarily the application of an existing modeling framework to significant applications.

---

> ### Author Response · Authors · 2024-11-25
> **General updates to the manuscript**
>
> Thank you for your comprehensive review of the manuscript. This has helped us significantly improve the content of the theoretical approach and experimentation. Here are the general updates we made to the manuscript:
>
> 1. Improved the Physics Informed Auto Encoder architecture to be consistent with the components of the SDE formalized in section 3.2 of the original manuscript. We have added another decoder that probabilistically predicts the noise term in the SDE i.e., $\sigma_t dW_t$ which is added to the decoder output for $NEE_{t+1}$ to generate the final value of $NEE_{t+1}$. This is done similar to how Variational Auto Encoder construct the probabilistic latent space with predicted values of $\mu$ and $log(\sigma^2)$. In our case, we sample a single value for noise term $\sigma_t dW_t$ using the noise decoder outputs of $\mu_{noise}$ and $log(\sigma_{noise}^2)$. According to White and Luo (2008) (referred to in line 168), for NEE modelling using SDEs, this noise term is same for both $dNEE_t$ and $NEE_t$, and is therefore also added to the drift $\mu_t dt$ in the SDE $\mathcal{N}_t$ in the PIAE architecture. This has led to a novel architecture for incorporating SDEs of this kind into Physics Informed Auto Encoder setups.
>
> 2. We analyzed the error distribution between the Physics Model defined in section 3.1 and the observed NEE values in the data. Based on the normality test, we show that this distribution is Gaussian and, therefore, can be used as ground truth for the noise in the data. The mean and standard deviation of this error distribution serve as the target distribution for our newly introduced noise decoder to converge towards.
>
> 3. Merged the two phases of the loss convergence to make them coherent based on the argument from Reviewer 4. We apply Maximum Mean Discrepancy (MMD) on the predicted stochastic NEE (with noise), while the Mean Squared Error (MSE) is used for all other decoder outputs (deterministic). The noise term is also aligned to the target error distribution (see point 2), using MMD. The training routine is now merged into a single phase with two weighted terms for both loss functions used.
>
> 4. Based on these improvements, we present significant improvements over original manuscript to all quoted metrics on the nighttime data. We include uncertainty analysis using standard errors for all quoted metrics (experimented over multiple runs of training and inference to cater for uncertainty resulting from model initialization and probabilistic predictions of noise term). For complete reproducibility, in the Appendix section, we add the complete experimentation, including all seed values used, hyperparameters, weight initialization technique and architecture specifications. The link to the Github repository containing the code is also included in the updated manuscript.
>
> 5. General improvements to the explanation (addressing reviewer remarks on SDE formalization, variables used etc.) and presentation of the concepts in the paper.
>
> We hope to share the updated manuscript before the stated deadline and look forward to your reviews.

---

> > ### Comment · Reviewer_mYoA · 2024-11-27
> >
> > Thank you for the response regarding general updates made to the manuscript. As the response does not directly address my concerns and a revised manuscript is not provided, I maintain my score for now.

---

> ### Author Response · Authors · 2024-11-29
> **Response after corrections, for Reviewer mYoA**
>
> Thank you for your answer. This answer to your review is related to the comment ‘Final (general) updates of the manuscript’ associated to the final version of the manuscript (on top).
>
> > Why is latent heat (L) excluded? Having a high correlation to the target variable would seem to be a good thing when the goal is to predict missing values?
>
> NEE and latent heat are measured by the same instrument. Thus, there is a high probability that latent heat is missing when NEE is missing. We put at the line 105-107 in the updated version of the manuscript.
>
> > The notation in equation 4-5 is difficult to read. Would it not be more clear to write this in terms of partial derivatives?
>
> Thank you for noticing this, we put the notation in terms of partial derivatives at line 179-186.
>
> > For improved readability, consider to use italics for variables and roman (upright) type for named functions, as subscripts in equations, and for units of measurement. Consider that multi-letter abbreviations can be confusing in equations: For example, it can be unclear if rb is a single variable or the product of r and b.
>
> We put the variables in italic, the functions, the subscripts in the equations and the units in roman upright. This makes notations clearer for the readers not familiar with the NEE model. We also replaced rb by r, for respiration.
>
> > rb (night/day) is not defined in the main text as far as I can see. rb is mentioned in the text in the appendix but not in the mathematical derivations.
>
> The base respiration for night and day are defined after the definition of the ecosystem respiration R_eco in section 3.1.1. and 3.1.2. It is not mentioned in the appendix. R_g, the global radiation is mentioned in the appendix. Is the reviewer referring to something else?
>
> > In equation 9, should is there not a difference between dt on the left and right hand side? On the left hand side, it seems to denote an infinitesimal element, and on the right side it is 30 minutes?
>
> Yes, thank you for pointing this out. The right part of equation 8, at line 230-231, is approximation. Thus, we referred to $\Delta t$ for the 30min time interval of measurement.
>
> > I am not sure how this approach is an autoencoder. As I understand the written description, the model predicts one timestep ahead with a latent encoding, and thus does forecasting rather than reconstruction. However, Figure 1 does seem to imply that the decoders predict for the same timestep.
>
> We tried to clarify the architecture scheme in figure 1. The decoder for $NEE_t$ does provide $NEE_{t+1}$. In the updated architecture, shown in Figure 1, the decoders reconstruct parts of the input space i.e., $NEE_t$, $k_t$, ... We do not reconstruct the complete the entries $X_t$ since that is not the goal of the work.
>
> > Is there something wrong with the linebreaks in Algorithm 1, step 4?
>
> Since there is already the line numbering of the paper, we removed the line numbers of the algorithm. Removing the comma at line 334 could help to clarify the reading of the algorithm, we are constrained by the line continuation of the equation at line 335-337.
>
> > What is the reason for the choice of the two loss phases?
>
> We have now merged the MSE and the MMD. The reasoning behind this in the loss function section is explained in section 4.5. for the loss function. We use early stopping to determine the end of convergence when the loss function plateaus. We use reduce learning rate on plateau technique with a patience value of 20 epochs, allowing the learning rate to fall till a minimum value of 0.00001 and stopping the training upon plateauing at this learning rate.
> https://pytorch.org/docs/stable/generated/torch.optim.lr_scheduler.ReduceLROnPlateau.html
>
> > I am not familiar with the literature on physics-informed autoencoders, but I would like to ask whether this paper introduces any technical contributions to the framework itself, or if the contribution is primarily the application of an existing modeling framework to significant applications.
>
> After the manuscript update, the novelty lies in the direct integration of the SDE into the PIAE architecture, and in its ability to forecast NEE as a combination of a deterministic drift and a noise term. This adds interpretability to the way the PIAE models SDE problems similar to the NEE one we present, as mentioned in 2. of the comment ‘Final (general) updates of the manuscript’.

---

> > ### Comment · Reviewer_mYoA · 2024-12-02
> >
> > Thank you for the detailed response. Improvements to the notation etc. have made the paper significantly easier to follow. I have updated my score to 5. I still recommend rejection for the reasons outlined in my initial review.

---

> > > ### Author Response · Authors · 2024-12-02
> > > **Answer to Official Comment by Reviewer mYoA**
> > >
> > > Thank you for your quick answer and your update to the score of our work.
> > >
> > > We updated the approach to add novelty. The novelty now lies in the introduction of PIAE for SDE with a stochastic sampler. This approach can be extended to other applications where we can integrate SDEs of this form into autoencoders by predicting both the drift term and the noise term associated with the physical process being modelled. This also adds interpretability to the model and therefore more applicability and benefit for the wider ICLR community.
> > >
> > > We are looking forward to your expert opinion on this approach.

---

### Official Review · Reviewer_Lwtz · 2024-11-03

**Soundness:** 3
**Presentation:** 1
**Contribution:** 3
**Rating:** 5
**Confidence:** 2

**Summary:**

The paper studies the application of autoencoders for the problem of imputing missing values in Co2 Net Ecosystem Exchange (NEE) measurements. The autoencoder takes in several covariates, such as temperatures and radiations, at a given timestep $t$ and predicts the next-step NEE, along with several variables of a Stochastic Differential Equation that models changes in NEE.

**Strengths:**

1. The paper applies ML techniques to enhance NEE measurements, which has the potential to improve the estimation of Co2 emissions, resulting in reduced uncertainty in our projections. **This is an important problem with high societal and environmental impact.**

**Weaknesses:**

1. **The presentation of the paper is convoluted, and requires a degree of familiarity with the NEE problem that is uncommon in the ICLR community**. It is then hard for me to judge the significance, originality and potential impact of the work.
 More precisely, these are some points that are not sufficiently explained or that make the paper hard to read and understand:
- In the introduction, it is mentioned that missing NEE values are due to e.g. power shortages. I assume that in such scenarios, the values of the covariates (temperature, radiation, etc) are also missing due to the same issue. However, the proposed model requires having access to all covariate values at a given time. How can the model be applied in practice without these values?
- In the introduction, the first highlighted contribution is the introduction of a SDE for NEE measurements. Put it that way, it sounds like the SDE is novel also in the physics. However this point is not stressed again later on, so I wonder whether the SDE is known and the novelty is in its use as supervision for learning ML models.
- Line 157, it is mentioned that the $E_0$ parameter is estimated with the nighttime model and used in the daytime one, but it is not explained why.
- In section 4.4, it is mentioned that the integration of the SDE in the training of the autoencoder follows previous work [Raissi 2017], but it is not sufficiently described to make the paper self-contained.
- The related work is not sufficiently described. In particular, it is not clear whether the reported baselines RFR and XgBoost variant based on the work of [Moffat 2007] are also physics-informed or only statistical.
- Second and third lines of Equation 5: do the second (from the left) commas separate two different definitions or do they indicate the continuation of the variable suffices?

2. The tables do not report standard errors, which makes impossible to judge the significance of the improvements.

3. The paper does not discuss limitations nor future work.

**Questions:**

1. Could the work be applied to other physical systems? Would that require knowledge of the DFE governing the system?

---

> ### Author Response · Authors · 2024-11-13
> **Initial Response**
>
> Thank you for sharing your comprehensive review on the manuscript. We are working on addressing each comment and improving the manuscript accordingly. We will share the updates with you soon.
>
> Meanwhile, we have a question on one of the comments:
>
> _“In section 4.4, it is mentioned that the integration of the SDE in the training of the autoencoder follows previous work [Raissi 2017], but it is not sufficiently described to make the paper self-contained.”_
>
> The SDE is directly made part of the neural architecture, with the mathematical operators in the SDE are direct nodes in the computation graph. We can view this as a non-trainable physics layer in the neural network architecture. Should we reflect this better in Figure 1?

---

> > ### Comment · Reviewer_Lwtz · 2024-11-25
> > **SDE integration**
> >
> > Thank you for describing the integration of the SDE in the architecture. This description should be reported in the main text. About Figure 1, it wouldn't hurt to update it accordingly.

---

> > ### Author Response · Authors · 2024-11-25
> > **General updates to the manuscript**
> >
> > Thank you for your comprehensive review of the manuscript. This has helped us significantly improve the content of the theoretical approach and experimentation. Here are the general updates we made to the manuscript:
> >
> > 1. Improved the Physics Informed Auto Encoder architecture to be consistent with the components of the SDE formalized in section 3.2 of the original manuscript. We have added another decoder that probabilistically predicts the noise term in the SDE i.e., $\sigma_t dW_t$ which is added to the decoder output for $NEE_{t+1}$ to generate the final value of $NEE_{t+1}$. This is done similar to how Variational Auto Encoder construct the probabilistic latent space with predicted values of $\mu$ and $log(\sigma^2)$. In our case, we sample a single value for noise term $\sigma_t dW_t$ using the noise decoder outputs of $\mu_{noise}$ and $log(\sigma_{noise}^2)$. According to White and Luo (2008) (referred to in line 168), for NEE modelling using SDEs, this noise term is same for both $dNEE_t$ and $NEE_t$, and is therefore also added to the drift $\mu_t dt$ in the SDE $\mathcal{N}_t$ in the PIAE architecture. This has led to a novel architecture for incorporating SDEs of this kind into Physics Informed Auto Encoder setups.
> >
> > 2. We analyzed the error distribution between the Physics Model defined in section 3.1 and the observed NEE values in the data. Based on the normality test, we show that this distribution is Gaussian and, therefore, can be used as ground truth for the noise in the data. The mean and standard deviation of this error distribution serve as the target distribution for our newly introduced noise decoder to converge towards.
> >
> > 3. Merged the two phases of the loss convergence to make them coherent based on the argument from Reviewer 4. We apply Maximum Mean Discrepancy (MMD) on the predicted stochastic NEE (with noise), while the Mean Squared Error (MSE) is used for all other decoder outputs (deterministic). The noise term is also aligned to the target error distribution (see point 2), using MMD. The training routine is now merged into a single phase with two weighted terms for both loss functions used.
> >
> > 4. Based on these improvements, we present significant improvements over original manuscript to all quoted metrics on the nighttime data. We include uncertainty analysis using standard errors for all quoted metrics (experimented over multiple runs of training and inference to cater for uncertainty resulting from model initialization and probabilistic predictions of noise term). For complete reproducibility, in the Appendix section, we add the complete experimentation, including all seed values used, hyperparameters, weight initialization technique and architecture specifications. The link to the Github repository containing the code is also included in the updated manuscript.
> >
> > 5. General improvements to the explanation (addressing reviewer remarks on SDE formalization, variables used etc.) and presentation of the concepts in the paper.
> >
> > We hope to share the updated manuscript before the stated deadline and look forward to your reviews.

---

> ### Author Response · Authors · 2024-11-29
> **Response after corrections, for Reviewer Lwtz**
>
> Thank you for your review. Please find here an answer to your review related to the comment ‘Final (general) updates of the manuscript’ associated to the final version of the manuscript (on top).
>
> > In the introduction, it is mentioned that missing NEE values are due to e.g. power shortages. I assume that in such scenarios, the values of the covariates (temperature, radiation, etc) are also missing due to the same issue. However, the proposed model requires having access to all covariate values at a given time. How can the model be applied in practice without these values?
>
> Remote sensing instruments can be used, such as satellite observations, to measure covariate values. This is mentioned in line 43-44.
>
> > In the introduction, the first highlighted contribution is the introduction of a SDE for NEE measurements. Put it that way, it sounds like the SDE is novel also in the physics. However this point is not stressed again later on, so I wonder whether the SDE is known and the novelty is in its use as supervision for learning ML models.
>
> Considering the drift part from Lasslop et al. 2010. with Gaussian noise is a novelty introduced by the paper. It is using 5 parameters, instead of 7 in White & Luo 2008. We highlighted this in the summary of part 3, at the line 117-118. The introduced architecture in Figure 1 is adapted to this
>
> > Line 157, it is mentioned that the $E_0$ parameter is estimated with the nighttime model and used in the daytime one, but it is not explained why.
>
> A major part of the ecosystem respiration R_eco is due to soil respiration which continues from night to day, $E_0$ being a major factor in ecosystem respiration it remains the same for night and day. This is stated at line 159-161.
>
> > In section 4.4, it is mentioned that the integration of the SDE in the training of the autoencoder follows previous work [Raissi 2017], but it is not sufficiently described to make the paper self-contained.
>
> We corrected the architecture in Figure 1 to show the decoder outputs being used as input to $\mathcal{N}_t$ (the drift component of the SDE).
>
> In the updated manuscript, we improve the description of the SDE as a combination of drift and noise terms from the perspective of the PIAE architecture in Figure 1. In Equation 6, we describe $\mathcal{N}_t$ as the drift term based on the operators in Equation 5. These operators are an explicit part of the computational graph of the PIAE neural network.
>
> > The related work is not sufficiently described. In particular, it is not clear whether the reported baselines RFR and XgBoost variant based on the work of [Moffat 2007] are also physics-informed or only statistical.
>
> RFR and XGBoost are not physics-informed, we talked about ‘conventional’ methods at line 367-368. RFR is statistical and XGBoost is a distributed gradient-boosted decision tree (GBDT) machine learning library.
>
> > Second and third lines of Equation 5: do the second (from the left) commas separate two different definitions or do they indicate the continuation of the variable suffices?
>
> We put the equation 5 in term of partial derivatives which makes things easier to understand.
>
> > The tables do not report standard errors, which makes impossible to judge the significance of the improvements.
>
> We updated the table 2 and included the standard deviations.
>
> > The paper does not discuss limitations nor future work.
>
> The work is limited to the stated SDE. Too rapid changes in the drift or non-Gaussian noise is a challenge for the PIAE model. Experimentalists tend to do additional measurements to ensure that to have more data points to avoid these behaviour of the data. Moreover, Extreme events (droughts, floods, heat wave, ...) impact on NEE could be studied with the PIAE, this expected by the geoscience community. We did not have the time to put this in the updated version of the manuscript. It can be added to the Camera ready version if necessary.
>
> > Could the work be applied to other physical systems? Would that require knowledge of the DFE governing the system?
>
> Yes, of course. This requires the knowledge of the dynamics described by a differential equation. If the physical systems differential equation is well defined, which is often the case, application to other domain are possible.

---

### Official Review · Reviewer_1KBR · 2024-11-04

**Soundness:** 3
**Presentation:** 3
**Contribution:** 3
**Rating:** 6
**Confidence:** 3

**Summary:**

This paper proposes Physics-Informed Autoencoders (PIAEs) to address gaps in CO2 emission measurements from agricultural fields. The method combines autoencoder architectures with physical Net Ecosystem Exchange (NEE) models, integrating equations that describe CO2 exchanges between the atmosphere and carbon pools (i.e., utilizing
the SDE defined as a Wiener process). Their main contribution is extending standard autoencoders with a stochastic differential equation framework that models NEE changes over time, particularly addressing nighttime measurement gaps. Their method also provides forecasting capabilities and enhances performance on NEE gap-filling by accurately learning the NEE distribution and associated parameters. They evaluate their approach on 8 years of flux tower data from East Anglia, showing improvements over current state-of-the-art methods, especially for nighttime predictions, where they achieve a 22% higher R2 score than Random Forest approaches.

**Strengths:**

1. Introducing a stochastic differential equation for NEE measurements combining daytime
and nighttime models with Gaussian noise.
2. Demonstrating that PIAE improves gap-filling robustness compared to state-of-the-art
methods, handling gaps from months to years.
3. Better Maximum Mean Discrepancy (MMD), Wasserstein distance, and Kullback-Leibler (KL) divergence validated significant improvements in NEE distribution learning.
4. Achieving better fit to NEE measurements validated by lower MAE and higher R2 scores.
5. Accurately predicting SDE parameters, enhancing interpretability.
6. Consistent improvement in nighttime predictions across metrics
7. Strong performance on distribution-based measures (MMD, Wasserstein, KL)
8. Ability to capture unusual events (e.g., downward NEE spikes)
9. Effective parameter estimation for both day and night models

**Weaknesses:**

1. While the supplementary material adequately explains the SDE derivation and diffusion coefficient determination, key points should be summarized in the main text. A brief note about how σnight and σday are derived from empirical error distributions would help readers understand the transition from Eq. 5 to 6 without requiring supplementary material consultation
2. AE is better than PIAE for all model parameter estimation across all metrics, contrary to their claim that their method enhances performance on NEE gap-filling by accurately learning the NEE distribution and associated parameters.
3. The computational requirements compared to simpler approaches like RF are not discussed.
4. The two-phase training procedure (MSE then MMD) has no convergence guarantees.
5. The claimed 22% improvement in R2 score lacks context - no variance was reported (Error bars or confidence intervals for the reported metrics would help). The hyperparameter selection process for PIAE and baseline models (including random forest) is not described. A fair comparison requires careful tuning of all methods.
6. Missing critical details:
    1. How were hyperparameters selected for PIAE and baselines?
    2. What are the network architectures (layer sizes, activation functions)?
    3. Where are the error bars and statistical significance tests?
    4. How does computational cost compare to simpler methods
7. The implementation details are insufficient for the reproduction
8. The comparisons in Figures 2 and 3 show selective periods without justification for their choice

Minor comments:
1. Section 4.5's description of the loss function uses inconsistent notation compared to earlier sections.
2. There are some writing clarity issues, like in lines 50 and 98.
3. The paper shows results across different timescales but doesn't systematically evaluate performance as a function of gap length. This would be valuable for understanding the method's practical utility.
4. The NEE parameter estimation details might fit better in methods

**Questions:**

1. The SDE formulation in Section 3.2 assumes specific forms for the drift and diffusion terms. The justification for these choices comes from prior work, but the implications of these modeling choices should be discussed. What happens when these assumptions are violated?
2. The two-phase training procedure using MSE then MMD requires more theoretical grounding:
    1. Why this specific sequence? How is convergence of the first phase determined before switching to MMD?
    2. Were other training strategies considered?
3. The choice of MMD kernels isn't discussed - how sensitive is the method to this choice?
4. How sensitive is the model to SDE parameter initialization?
5. What's the computational overhead versus RF/XGBoost?

---

> ### Author Response · Authors · 2024-11-13
> **Initial Response**
>
> Thank you for sharing your comprehensive review on the manuscript. We are working on addressing each comment and improving the manuscript accordingly. We will share the updates with you soon.
>
> Meanwhile, we have a question on one of the comments:
> _“The SDE formulation in Section 3.2 assumes specific forms for the drift and diffusion terms. The justification for these choices comes from prior work, but the implications of these modelling choices should be discussed. What happens when these assumptions are violated?”_
>
> Our formalization of NEE in terms of an SDE is based on the two papers referred to in line 165 (White & Luo 2008; Weng (2011)), with a drift term and a noise term. According to both of them, the noise term can be defined as a Gaussian process.  The drift equations in Equation 5 are derived from the model defined in Equations 1,2 and 3. The parameters inside the drift term are predicted by the decoders in the main PIAE model, described in section 4 later.
> Is your question specifically around the parameters σnight and σday in the noise term and the drift? Do you mean we need to do more precise study on each term? Are you referring to the state-of-the-art assumptions?

---

> > ### Author Response · Authors · 2024-11-25
> > **General updates to the manuscript**
> >
> > Thank you for your comprehensive review of the manuscript. This has helped us significantly improve the content of the theoretical approach and experimentation. Here are the general updates we made to the manuscript:
> >
> > 1. Improved the Physics Informed Auto Encoder architecture to be consistent with the components of the SDE formalized in section 3.2 of the original manuscript. We have added another decoder that probabilistically predicts the noise term in the SDE i.e., $\sigma_t dW_t$ which is added to the decoder output for $NEE_{t+1}$ to generate the final value of $NEE_{t+1}$. This is done similar to how Variational Auto Encoder construct the probabilistic latent space with predicted values of $\mu$ and $log(\sigma^2)$. In our case, we sample a single value for noise term $\sigma_t dW_t$ using the noise decoder outputs of $\mu_{noise}$ and $log(\sigma_{noise}^2)$. According to White and Luo (2008) (referred to in line 168), for NEE modelling using SDEs, this noise term is same for both $dNEE_t$ and $NEE_t$, and is therefore also added to the drift $\mu_t dt$ in the SDE $\mathcal{N}_t$ in the PIAE architecture. This has led to a novel architecture for incorporating SDEs of this kind into Physics Informed Auto Encoder setups.
> >
> > 2. We analyzed the error distribution between the Physics Model defined in section 3.1 and the observed NEE values in the data. Based on the normality test, we show that this distribution is Gaussian and, therefore, can be used as ground truth for the noise in the data. The mean and standard deviation of this error distribution serve as the target distribution for our newly introduced noise decoder to converge towards.
> >
> > 3. Merged the two phases of the loss convergence to make them coherent based on the argument from Reviewer 4. We apply Maximum Mean Discrepancy (MMD) on the predicted stochastic NEE (with noise), while the Mean Squared Error (MSE) is used for all other decoder outputs (deterministic). The noise term is also aligned to the target error distribution (see point 2), using MMD. The training routine is now merged into a single phase with two weighted terms for both loss functions used.
> >
> > 4. Based on these improvements, we present significant improvements over original manuscript to all quoted metrics on the nighttime data. We include uncertainty analysis using standard errors for all quoted metrics (experimented over multiple runs of training and inference to cater for uncertainty resulting from model initialization and probabilistic predictions of noise term). For complete reproducibility, in the Appendix section, we add the complete experimentation, including all seed values used, hyperparameters, weight initialization technique and architecture specifications. The link to the Github repository containing the code is also included in the updated manuscript.
> >
> > 5. General improvements to the explanation (addressing reviewer remarks on SDE formalization, variables used etc.) and presentation of the concepts in the paper.
> >
> > We hope to share the updated manuscript before the stated deadline and look forward to your reviews.

---

> > > ### Comment · Reviewer_1KBR · 2024-11-26
> > >
> > > Thank you for asking for clarification about the SDE formulation comment. The concern was not about the fundamental choice of using a Wiener process or Gaussian noise terms, which are well-justified by the cited works, but rather about understanding how the model performs when real-world conditions deviate from the ideal mathematical assumptions. A brief discussion of model behavior under non-ideal conditions (e.g., non-Gaussian noise, rapid changes in drift terms) would strengthen the paper and help readers understand its practical limitations.

---

> > > > ### Author Response · Authors · 2024-12-02
> > > > **Response after corrections, for Reviewer 1KBR, weaknesses**
> > > >
> > > > Thank you for your review. Please find here an answer to your review related to the comment ‘Final (general) updates of the manuscript’ associated to the final version of the manuscript (on top).
> > > >
> > > > > W 1. While the supplementary material adequately explains the SDE derivation and diffusion coefficient determination, key points should be summarized in the main text. A brief note about how σnight and σday are derived from empirical error distributions would help readers understand the transition from Eq. 5 to 6 without requiring supplementary material consultation
> > > >
> > > > We now added a brief note in the description of the SDE equation 4. section 3.2
> > > >
> > > > > W 2. AE is better than PIAE for all model parameter estimation across all metrics, contrary to their claim that their method enhances performance on NEE gap-filling by accurately learning the NEE distribution and associated parameters.
> > > >
> > > > We wanted to make a comparison in the reconstruction of the parameters of the SDE by the PIAE. AE does not use the parameters in the SDE to reconstruct NEE. Thus, we moved that in the supplementary material.
> > > >
> > > > > W 3. The computational requirements compared to simpler approaches like RF are not discussed.
> > > >
> > > > The inference time cost of PIAE, RFR and XGBoost are in the same order of magnitude on our machine, approximately few seconds. Training time is approximately 20min for our method, and 1min for RFR and XGBoost. This is briefly mentioned at line 369, in section 5. If required, a more refined study can be added to the supplementary material in the Camera-ready version, if the paper is accepted.
> > > >
> > > > > W 4. The two-phase training procedure (MSE then MMD) has no convergence guarantees.
> > > >
> > > > We have now merged the MSE and the MMD. The reasoning behind this in the loss function section is explained in section 4.5. for the loss function. We use early stopping to determine the end of convergence when the loss function plateaus. We use reduce learning rate on plateau technique with a patience value of 20 epochs, allowing the learning rate to fall till a minimum value of 0.00001 and stopping the training upon plateauing at this learning rate.
> > > > https://pytorch.org/docs/stable/generated/torch.optim.lr_scheduler.ReduceLROnPlateau.html
> > > >
> > > > > W 5. The claimed 22% improvement in R2 score lacks context - no variance was reported (Error bars or confidence intervals for the reported metrics would help). The hyperparameter selection process for PIAE and baseline models (including random forest) is not described. A fair comparison requires careful tuning of all methods.
> > > >
> > > > We now reported the new MAE and R2 score with standard deviation in table 2. We chose the initial hyperameters based on empirical analysis, we ran five training experiments for each method with seed values 0, 51, 123, 255, 999 to cater for the randomness/uncertainty resulting from the training and inference.
> > > >
> > > > > W 6. 1. How were hyperparameters selected for PIAE and baselines?
> > > >
> > > > cf. W 5. for hyperparameters. RFR comes from [Zhu et al. 2022] as mentioned in the paper in line 52-53, XGBoost is a distributed gradient-boosted decision tree (GBDT) machine learning library which can be a comparison point with RFR. AE is introduced to show the improvement done by the Physics model used in PIAE.
> > > >
> > > > > W 6. 2. What are the network architectures (layer sizes, activation functions)?
> > > >
> > > > The network architecture is given in figure 4 and 5 of the appendix. It is also available in the code https://github.com/saadzia10/PIAE-SDE which can be added in the Camera-ready version, if the paper is accepted.
> > > >
> > > > > W 6. 3. Where are the error bars and statistical significance tests?
> > > >
> > > > The 96% confidence intervals were added to the plots in figure 2 as shaded regions.
> > > >
> > > > > W. 6. 4. How does computational cost compare to simpler methods?
> > > >
> > > > Please, see  W 3.
> > > >
> > > > > W 7. The implementation details are insufficient for the reproduction
> > > >
> > > > We have included the complete network architecture in figure 4. and 5. in the appendix. We also provided the code now cf. W 6.2.
> > > >
> > > > > W 8. The comparisons in Figures 2 and 3 show selective periods without justification for their choice

---

> > > > > ### Author Response · Authors · 2024-12-02
> > > > > **Response after corrections, for Reviewer 1KBR, weaknesses, minor comments**
> > > > >
> > > > > Thank you for these minor comments. You will find the answers to these remarks here.
> > > > >
> > > > > > W. Minor comments : Section 4.5's description of the loss function uses inconsistent notation compared to earlier sections.
> > > > >
> > > > > We represent randomly selected time window from the test set to show that the trend is well captured by the algorithm. We can add more results on the test set in the appendix in the Camera-ready version if accepted.
> > > > >
> > > > > > W. Minor comments : There are some writing clarity issues, like in lines 50 and 98.
> > > > >
> > > > > We updated the loss function description in equation 16, section 4.5.
> > > > >
> > > > > > W. Minor comments : The paper shows results across different timescales but doesn't systematically evaluate performance as a function of gap length. This would be valuable for understanding the method's practical utility.
> > > > >
> > > > > We put R2 at line 52 in the new manuscript. We tried to make the line 104-106 clearer.
> > > > >
> > > > > > W. Minor comments : The NEE parameter estimation details might fit better in methods
> > > > >
> > > > >  Is this an incomplete comment by mistake? Can you please elaborate?

---

> > > > > > ### Author Response · Authors · 2024-12-02
> > > > > > **Response after corrections, for Reviewer 1KBR, questions**
> > > > > >
> > > > > > Thank you for your questions. You will find our answers here.
> > > > > >
> > > > > > > Q 1. The SDE formulation in Section 3.2 assumes specific forms for the drift and diffusion terms. The justification for these choices comes from prior work, but the implications of these modeling choices should be discussed. What happens when these assumptions are violated?
> > > > > >
> > > > > > In practice, when new parameters are fitted every few days, then this is to avoid these issues with situations where the observations change quickly and become non-homogeneous because the underlying ecosystem changes.
> > > > > >
> > > > > > Similarly, it tries to avoid non-gaussian noise, that could arise from a change in land management or particular weather conditions.
> > > > > > Thus, the PIAE is limited to this framework. We can add that to the limitations in the Camera-ready version if the paper is accepted.
> > > > > >
> > > > > > > Q 2. The two-phase training procedure using MSE then MMD requires more theoretical grounding:
> > > > > > > Why this specific sequence? How is convergence of the first phase determined before switching to MMD?
> > > > > > > Were other training strategies considered?
> > > > > >
> > > > > > We merged MSE and MMD. We refer to W 4. of the weaknesses (see above).
> > > > > >
> > > > > > > The choice of MMD kernels isn't discussed - how sensitive is the method to this choice?
> > > > > >
> > > > > > We have chosen Gaussian kernel as in Zhong and Meidani 2023 [1]. As stated in Zhong & Meidani 2023 [1] and Gretton et al. [2], the evaluation of MMD with various characteristic kernels have been shown to be consistent.
> > > > > >
> > > > > > > How sensitive is the model to SDE parameter initialization?
> > > > > >
> > > > > > For the scope of this paper, we did not include this study. But, this is a very insightful remark and we will include it in our future work.
> > > > > >
> > > > > > > What's the computational overhead versus RF/XGBoost?
> > > > > >
> > > > > > Please, see W 3. above
> > > > > >
> > > > > > **References**
> > > > > >
> > > > > > [1] PI-VAE: Physics-Informed Variational Auto-Encoder for stochastic differential equations, W. Zhong and H. Meidani , Computer Methods in Applied Mechanics and Engineering, Volume 403, Part A, 2023, https://doi.org/10.1016/j.cma.2022.115664.
> > > > > >
> > > > > > [2] A kernel two-sample set, Gretton A., Borgwardt K.M., Rasch M.J., Schölkopf B., Smola A., Journal of Machine Learning Research, 13, pp. 723 - 773, 2012, https://www.scopus.com/inward/record.uri?eid=2-s2.0-84859477054&partnerID=40&md5=f97ffe4fcf56556ac7c5f2822d03a841

---

### Author Response · Authors · 2024-11-28
**Final (general) updates in the revised manuscript**

We are grateful for the respected reviewers' comprehensive review of the manuscript. This has helped us significantly improve the theoretical approach and experimentation content. We now present the final updates to the manuscript:

1. We updated the use of the original SDE: $d \ \text{NEE}_t = \upmu_t dt + \upsigma_t d \text{W}_t$ composed of deterministic drift and noise terms to be consistent in the PIAE architecture. Here, the PIAE now outputs both a drift term and a probabilistic noise term as part of its outputs as shown in Equation 6 in the updated manuscript: $d \ \text{NEE}_t =\text{f}_t(\omega) +\varepsilon_t(\omega), \omega \in \Omega$, where $\text{f}_t$ is the drift (also known as forcing term) and $\varepsilon_t$ the noise term respectively.

2. The PIAE architecture now comprises six decoders (previously five) to reconstruct variables used as SDE components, NEE at the current time instance $t$ and the noise term. The predicted variables and the noise term are fed to $\mathcal{N}_t$ to compute the change in NEE (drift term): $\text{f}_t$, as shown in Equation 11. This is then added to the reconstructed NEE to forecast the NEE for time instance $t+1$, as shown in Equation 13 in the updated manuscript. The decoder for the noise term $\varepsilon_t$ predicts the mean and log variance of the noise, which are then used to sample a noise value using the reparameterization trick as done in Variational Auto Encoders. This is highlighted in Figure 1.

3. We merged the two phases of the loss convergence to make them coherent based on the argument from Reviewer 4. The loss term comprises a weighted sum of two cost functions: Maximum Mean Discrepancy (MMD) and Mean Squared Error (MSE). MMD is now only applied to the predicted stochastic $\text{NEE}_{t+1}$ (which has the noise component added to it) and to align the predicted noise term $\varepsilon_t$ to the target error distribution (see point 4 below and Equations 15, 16 in manuscript). MSE is used for all other decoder outputs (deterministic).

4. We analyzed the error distribution between the Physics Model defined in section 3.1 and the observed NEE values in the data. This error distribution's mean and standard deviation serve as the target distribution for our newly introduced noise decoder to converge towards.

5. Based on these improvements, we present improvements to the quoted distribution-based metrics (MMD, Wassertein Distance and KL Divergence) on the nighttime data. We include uncertainty analysis using standard errors for all quoted metrics (experimented over multiple runs of training and inference to cater for uncertainty resulting from model initialization and probabilistic predictions of noise term). We also present the neural network architecture configuration (layers and parameters) for both PIAE and AE and the configuration of the hyperparameters for the RF and XG Boost, in the Appendix section.

6. We forgot to add the Github repository link to the codebase. For reproducibility, we are providing the link below and hope to add this to the manuscript for the camera-ready version if accepted:
https://github.com/saadzia10/PIAE-SDE

7. General improvements to the explanation (addressing reviewer remarks on SDE formalization, variables used etc.) and presentation of the concepts in the paper.

Again, we thank the reviewers for their constructive feedback. We will now follow up with individual responses from each reviewer.

---

### Meta-Review · Area_Chair_SqF1 · 2024-12-18

**Metareview:**

The paper proposes Physics-Informed Autoencoders (PIAEs) to address gaps in carbon dioxide (CO2) emission measurements, specifically for Net Ecosystem Exchange (NEE) data from agricultural fields. The model combines machine learning with physical NEE models by integrating stochastic differential equations (SDEs) to enhance the accuracy and reliability of predictions, particularly at night, when data gaps are common. The authors employ a two-phase training process, optimizing for Mean Squared Error (MSE) initially and Maximum Mean Discrepancy (MMD) later, to improve imputation and forecasting of NEE values.  According to the reviews, the paper has weaknesses including limited novelty, as the work mainly applies existing methods to a specific domain. The paper has poor clarity in presenting mathematical formulations, design choices, and training methodology. There are missing details on computational costs, hyper-parameter tuning, and statistical significance of results.  There is a lack of broader applicability and justification for key design elements, such as the two-phase training process.  Summing up, while the results are promising, the paper's complexity and missing details limit its impact on a general machine-learning audience. It may be better suited for a domain-specific conference after revisions.

**Additional Comments On Reviewer Discussion:**

During the discussion, the authors provided feedback on several issues raised by the reviewers. However, in general, most of the reviewers' concerns about this paper remain after the discussion.

---

### Decision · Program_Chairs · 2025-01-22

Reject